# To smooth a cloud or to pin it down: Guarantees and Insights on Score Matching in Denoising Diffusion Models

Teodora Reu * 1          Francisco Vargas * 2          Anna Kerekes[2]          Michael M. Bronstein[1]

[1]University of Oxford,
[2]University of Cambridge,

## Abstract

Denoising diffusion models are a class of generative models that have recently achieved state-of-the-art results across many domains. Gradual noise is added to the data using a diffusion process, which transforms the data distribution into a Gaussian. Samples from the generative model are then obtained by simulating an approximation of the time reversal of this diffusion initialized by Gaussian samples. Recent research has explored the sampling error achieved by diffusion models under the assumption of an absolute error $\epsilon$ achieved via a neural approximation of the score. To the best of our knowledge, no work formally quantifies the error of such neural approximation to the score. In this paper, we close the gap and present quantitative error bounds for approximating the score of denoising diffusion models using neural networks leveraging ideas from stochastic control. Finally, through simulation, we explore some of the insights that arise from our results confirming that diffusion models based on the Ornstein-Uhlenbeck (OU) process require fewer parameters to better approximate the score than those based on the Fölmer drift / Pinned Brownian Motion.

## 1 INTRODUCTION

Let $\pi$ be a probability density on $\mathbb{R}^d$ of the form

$$\pi(x) = \frac{\gamma(x)}{Z}, \qquad Z = \int_{\mathbb{R}^d} \gamma(x)\mathrm{d}x, \qquad (1)$$

where $\gamma : \mathbb{R}^d \rightarrow \mathbb{R}^+$ can be evaluated pointwise but the normalizing constant $Z$ is intractable. In both the sampling problem and generative modelling, one is interested in obtaining approximate samples from $\pi$. In sampling, one has

---
*Equal contribution.

access to $\gamma$ whilst in generative modelling we only have access to samples from $x_i \sim \pi(x)$.

While superficially similar, methodologies for these two different tasks initially evolved quite separately. Due to the ability to take gradient sampling, a variety of Markov Chain Monte Carlo (MCMC) [Neal, 2011], as well as variational [Wainwright et al., 2008, Blei et al., 2017] techniques have been developed to tackle the sampling problem. In variational techniques, one considers a flexible family of easy-to-sample distributions $q^\theta$ whose parameters are optimized by minimizing a suitable cost, such as reverse Kullback–Leibler discrepancy $\mathrm{KL}(q^\theta || \pi)$.

Complementary, generative modelling is interested in being able to sample from the underlying density $\pi$ when only a set of finite samples is available. As a result, most methodologies were initially based on forward KL (i.e. Maximum Likelihood) like approaches, where one trains a tractable model $q^\theta$ via minimizing $\mathrm{KL}(\pi || q^\theta)$ [Papamakarios et al., 2021] which can be achieved as we can estimate gradients $\nabla_\theta \mathrm{KL}(\pi || q^\theta)$ using samples from $\pi$.

Recent score-based techniques for generative modelling [Song et al., 2021b] constitute of nice cross-pollination between the standard techniques used in sampling (e.g. MCMC) ported over to generative modelling and in some cases feeding back into the sampling community [Doucet et al., 2022, Vargas et al., 2023].

In recent years we have seen the rise of Denoising Diffusion Probabilistic Models (DDPM), a powerful class of generative models [Sohl-Dickstein et al., 2015, Ho et al., 2020, Song et al., 2021b] to sample from unnormalized densities. In this context, one adds noise progressively to data using diffusion to transform the complex target distribution into a Gaussian distribution. The time reversal of this diffusion can then be used to transform a Gaussian sample into a sample from the target. As with many theoretical works pertaining to diffusion models [Chen et al., 2022, Lee et al., 2023], we will assume the target distribution admits a density for our analysis; this is a common assumption in the

analysis of sampling algorithms [Ma et al., 2019, Vempala and Wibisono, 2019] and is not restrictive.

It is important to highlight that diffusion models have also recently made it into sampling [Vargas et al., 2023, Berner et al., 2022], in particular, these works establish connections between DDPM and well-established field of stochastic control [Kappen et al., 2012, Nüsken and Richter, 2021].

In this work, we delve into the connection to stochastic control highlighted among denoising diffusion models [Ho et al., 2020, Song et al., 2021b] in Vargas et al. [2023]. We leverage this connection to show how the score of VP-SDEs [Song et al., 2021b] can be approximated with neural networks up to an arbitrarily small error, and we quantify the induced sampling error.

Our contributions in this paper can be summarized as follows:

- Establishing a connection between the VP-SDE score and OU-semigroup (Section 3.2).

- Exploring novel regularity properties for OU-semigroup (Section 3.3), via leveraging connections to stochastic control [Tzen and Raginsky, 2019b].

- Demonstrating neural network and sampling approximation results for a simplified VP-SDE (Proposition 3.1, Remark 3.2) with minimal assumptions on the data/target distribution $\pi$.

- Leveraging some of the insights/conjectures motivated by our theoretical results we carry out a set of empirical explorations contrasting two different types of approaches of SDEs for score-based generative modelling (Föllmer Drift vs VP-SDE).

Upon comparing our approach with related works, notable differences in assumptions emerge. For instance, Chen et al. [2023] make manifold assumptions regarding data representation, diverging substantially from our less restrictive assumptions. Additionally, their methodology employs covers based on score networks rather than the OU-semigroup, which contrasts with ours. Similarly, in Oko et al. [2023], Cole and Lu [2024], assumptions about data distribution and analysis methods differ notably from ours, with their analysis focusing on score loss rather than the OU-semigroup. We highlight that whilst the concurrent work in Cole and Lu [2024] has similar assumptions on the target distribution, their focus is strictly on the data-driven case, and whilst some overlap in the spirit of the sketch, they obtain different bounds.

## 2 BACKGROUND - DENOISING DIFFUSION MODELS AND STOCHASTIC CONTROL

For this work, we will introduce Denoising Diffusions in continuous time. Let $\mathcal{C} = C([0, T], \mathbb{R}^d)$ be the space of continuous functions from $[0, T]$ to $\mathbb{R}^d$ and $\mathcal{B}(\mathcal{C})$ the Borel sets on $\mathcal{C}$. We consider path measures, which are probability measures on $(\mathcal{C}, \mathcal{B}(\mathcal{C}))$ [Léonard, 2014]. To relate denoising diffusion models to the methodology in Tzen and Raginsky [2019b], we will introduce connections presented in Vargas et al. [2023], which relate score matching in VP-SDEs to stochastic control, thus enabling our main results.

### 2.1 BACKWARDS DIFFUSION AND ITS TIME-REVERSAL

Consider the forward noising diffusion given by a time-reversed Ornstein–Uhlenbeck (OU) process (Song et al. [2021b] refer to this SDE as the VP-SDE):

$$\mathrm{d}x_t = -\beta_t x_t \mathrm{d}t + \sigma \sqrt{2\beta_t} \mathrm{d}B_t, \qquad x_0 \sim \pi, \quad (2)$$

where $(B_t)_{t \in [0,T]}$ is a $d$-dimension Brownian motion and $t \to \beta_t$ is a non-decreasing positive function. This diffusion induces the path-measure $\mathcal{P}$ on the time interval $[0, T]$, and the marginal density of $x_t$ is denoted $p_t$. The transition density of this diffusion is given by $p_{t|0}(x_t|x_0) = \mathcal{N}(x_t; \sqrt{1 - \lambda_t} x_0, \sigma^2 \lambda_t I)$, where $\lambda_t = 1 - \exp(-2 \int_0^t \beta_s \mathrm{d}s)$. We will always consider a scenario where $\int_0^T \beta_s \mathrm{d}s \gg 1$ so that $p_T(x) \approx \mathcal{N}(x; 0, \sigma^2 I)$.

From [Haussmann and Pardoux, 1986], its time-reversal $(y_t)_{t \in [0,T]} = (x_{T-t})_{t \in [0,T]}$, where equality is here in distribution, yields the forward time diffusion:

$$\mathrm{d}y_t = \beta_{T-t}\{y_t + 2\sigma^2 \nabla \ln p_{T-t}(y_t)\}\mathrm{d}t$$
$$+ \sigma \sqrt{2\beta_{T-t}} \mathrm{d}W_t, \quad y_0 \sim p_T, \quad (3)$$

where $(W_t)_{t \in [0,T]}$ is another $d$-dimensional Brownian motion. By definition this time-reversal starts from $y_0 \sim p_T = \mathrm{Law}(x_t) \approx \mathcal{N}(0, \sigma^2 I)$ and is such that $y_T \sim \pi$. This suggests that approximate simulation of diffusion (3) would result in approximate samples from $\pi$. However, putting this idea into practice requires being able to approximate the intractable scores $(\nabla \ln p_t(x))_{t \in [0,T]}$. Unlike DDPM, score matching techniques are not feasible, as sampling from (2) requires sampling $x_0 \sim \pi$, which is impossible by assumption.

### 2.2 REFERENCE DIFFUSION AND VALUE FUNCTION

To introduce the value function, it is first useful to introduce a *reference* process defined by the diffusion following

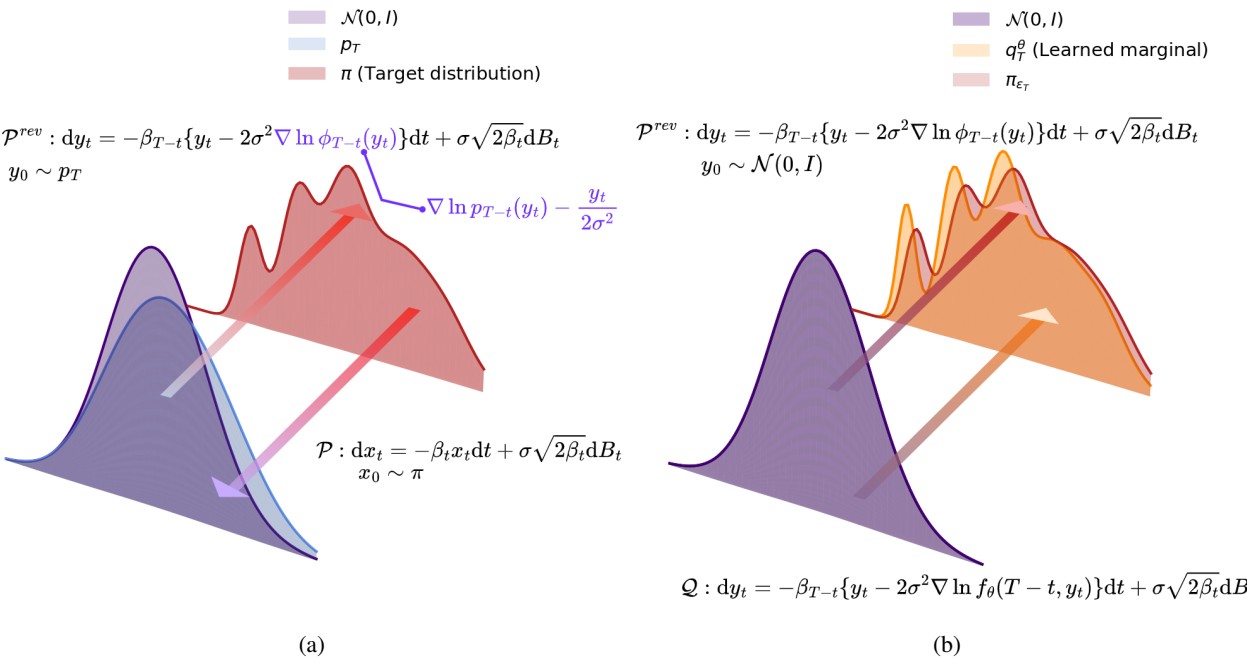

(a)

(b)

Figure 1: a) Noise-adding process for exact reversal. The distribution $\mathcal{N}(0, I)$ is drawn for comparison to $p_T$. b)Exact and approximate time reversal starting from $\mathcal{N}(0, I)$ the former exhibits only the mixing error whilst the latter incorporates the network's approximation error.

(2), but initialized at $p_0^{\text{ref}}(x) = \mathcal{N}(x; 0, \sigma^2 I)$ rather than $\pi(x)$ thus ensuring that the marginals of the resulting path measure $\mathcal{P}^{\text{ref}}$ all satisfy $p_t^{\text{ref}}(x) = \mathcal{N}(x; 0, \sigma^2 I)$. Following Vargas et al. [2023] we can identify $\mathcal{P}$ as the path measure minimizing the half bridge $\mathcal{P} = \arg\min_{\mathcal{Q}} \{\text{KL}(\mathcal{Q}||\mathcal{P}^{\text{ref}}) : q_T = \pi\}$ [Bernton et al., 2019, Vargas et al., 2021b, De Bortoli et al., 2021]. where representation of $\mathcal{P}^{\text{ref}}$ is given by

$$\mathrm{d}y_t = -\beta_{T-t}y_t\mathrm{d}t + \sigma\sqrt{2\beta_{T-t}}\mathrm{d}W_t, \qquad y_0 \sim p_0^{\text{ref}}. \quad (4)$$

Noting that $\beta_{T-t}(y_t + 2\sigma^2\nabla \ln p_t^{\text{ref}}(y_t)) = -\beta_{T-t}y_t$, Vargas et al. [2023] rewrites the time-reversal (3) of $\mathcal{P}$ as

$$\mathrm{d}y_t = -\beta_{T-t}\{y_t - 2\sigma^2\nabla \ln \phi_{T-t}(y_t)\}\mathrm{d}t$$
$$+ \sigma\sqrt{2\beta_{T-t}}\mathrm{d}W_t, \quad y_0 \sim p_T, \quad (5)$$

where $v_t(x) = -\ln\phi_t(x) = -\ln p_t(x)/p_t^{\text{ref}}(x)$ is known as the value function [Fleming and Rishel, 2012, Pham, 2009, Nüsken and Richter, 2021, Tzen and Raginsky, 2019b]. We point the reader to Figure 1 for a pictorial illustration of the aforementioned reversal and value function.

### 2.3 LEARNING THE FORWARD DIFFUSION - REVERSE KL / STOCHASTIC CONTROL FORMULATION

To approximate (3), consider a path measure $\mathcal{Q}^\theta$ which is induced by

$$\mathrm{d}y_t = \beta_{T-t}\{y_t + 2\sigma^2 s_\theta(t, y_t)\}\mathrm{d}t$$
$$+ \sigma\sqrt{2\beta_{T-t}}\mathrm{d}W_t, \quad y_0 \sim \mathcal{N}(0, \sigma^2 I), \quad (6)$$

so that $y_t \sim q_t^\theta$. To obtain $s_\theta(t, x) \approx \nabla \ln p_t(x)$, we parameterize $s_\theta(t, x)$ by a neural network whose parameters are obtained by minimizing

$$\text{KL}(\mathcal{Q}^\theta||\mathcal{P}) = \text{KL}(\mathcal{N}(0, \sigma^2 I)||p_T)$$
$$+\sigma^2\mathbb{E}_{\mathcal{Q}^\theta}\left[\int_0^T \beta_{T-t}||s_\theta(T-t, y_t)-\nabla \ln p_{T-t}(y_t)||^2\mathrm{d}t\right].$$

This expression closely resembles the expression obtained in Theorem 1 of Song et al. [2021a]. Note, in the DDPM Ho et al. [2020] setting as we have samples from $\mathcal{P}$ via simulating the forward SDE then one can recover the score-matching objective from Song et al. [2021b],

$$\text{KL}(\mathcal{P}||\mathcal{Q}^\theta) = \text{KL}(\mathcal{N}(0, \sigma^2 I)||p_T)$$
$$+\sigma^2\mathbb{E}_{\mathcal{P}}\left[\int_0^T \beta_{T-t}||s_\theta(T-t, y_t)-\nabla \ln p_{T-t}(y_t)||^2\mathrm{d}t\right].$$

To fully make the connection to stochastic control [Tzen and Raginsky, 2019b, Dai Pra, 1991], following Vargas et al. [2023] we equivalently reparameterize $\mathcal{Q}^\theta$ via the value function formulation of the backward SDE (Equation 5),

$$\mathrm{d}y_t = -\beta_{T-t}\{y_t - 2\sigma^2 f_\theta(T - t, y_t)\}\mathrm{d}t$$
$$+ \sigma\sqrt{2\beta_{T-t}}\mathrm{d}W_t, \qquad y_0 \sim \mathcal{N}(0, \sigma^2 I), \quad (7)$$

unlike Equation 6 $f_\theta$ approximates $\nabla \ln \phi_t$ rather than the score $\nabla \ln p_t$. Then under this reparameterization Vargas et al. [2023] use standard results on half bridges [Bernton et al., 2019] we can re-express $\mathrm{KL}(\mathcal{Q}^\theta || \mathcal{P})$ in the following form:

$$\mathrm{KL}(\mathcal{Q}^\theta || \mathcal{P}) = \mathbb{E}_{\mathcal{Q}^\theta}\left[ \sigma^2 \int_0^T \beta_{T-t} ||f_\theta(T - t, y_t)||^2 \mathrm{d}t \right.$$
$$\left. + \ln\left( \frac{\mathcal{N}(y_T; 0, \sigma^2 I)}{\pi(y_T)} \right) \right], \quad (8)$$

where $q_0^{\theta^*} = p_T \approx \mathcal{N}(0, \sigma^2 I)$ [1]. Then $\theta$ minimizing (8), approximate samples from $\pi$ can be obtained by simulating (7) and returning $y_T \sim q_T^\theta$. Note concurrent work [Berner et al., 2022] also optimizes an equivalent reverse KL to Equation 8. Equation 8 is an instance of stochastic control [Kappen et al., 2012, Tzen and Raginsky, 2019b, Nüsken and Richter, 2021, Berner et al., 2022] akin to the objective studied in [Tzen and Raginsky, 2019b], these re-formulations as a stochastic control problem, in particular, the connection to the value function (see Remark 3.5) will allow us to provide expresiveness remarks for VP-SDE based diffusions.

### 2.4   PINNED BROWNIAN MOTION GENERATIVE MODELS AND SAMPLERS

In this section, we reintroduce the class of generative models and samplers studied in [Tzen and Raginsky, 2019b] and highlight the similarities and differences in contrast to the OU-based diffusion models.

The pinned Brownian motion SDE is arrived at by using the h-transform to condition the scaled Brownian motion

$$\mathrm{d}x_t = \sqrt{\frac{\mathrm{d}\alpha_t}{\mathrm{d}t}}\mathrm{d}B_t, \qquad x_0 \sim \pi, \quad (9)$$

to hit the value 0 at time $T$, resulting in the forward SDE:

$$\mathrm{d}x_t = -\frac{\mathrm{d}\alpha_t}{\mathrm{d}t}\frac{x_t}{\alpha_T - \alpha_t}\mathrm{d}t + \sqrt{\frac{\mathrm{d}\alpha_t}{\mathrm{d}t}}\mathrm{d}B_t, \quad x_0 \sim \pi, \quad (10)$$

where $x_T = 0$, furthermore the SDE in Equation 10 has the following transition density (full derivation in Appendix F):

$$p(x_t|x_0) = \mathcal{N}\left( x_t \left| \frac{\alpha_T - \alpha_t}{\alpha_T - \alpha_0}x_0, \frac{(\alpha_T - \alpha_t)(\alpha_t - \alpha_0)}{\alpha_T - \alpha_0} \right. \right),$$

---

[1] $q_0^{\theta^*}$ denotes the optimal distribution at $t = 0$ minimising (8)

which we can use to learn the score [Song et al., 2021b]. Once we have the score the time reversal of Equation 10, yields an SDE which we can use for generative modelling

$$\mathrm{d}y_t = \frac{\mathrm{d}\alpha_{T-t}}{\mathrm{d}t}\left\{ \frac{y_t}{\alpha_T - \alpha_{T-t}} + \nabla \ln p_{T-t}(y_t) \right\}\mathrm{d}t$$
$$+ \sqrt{\frac{\mathrm{d}\alpha_{T-t}}{\mathrm{d}t}}\mathrm{d}W_t, \quad y_0 = 0. \quad (11)$$

We will refer to this SDE as the backward pinned brownian motion (BPBM). As we will discuss the BPBM SDE is a Generalisation of the Föllmer process [Dai Pra, 1991], which is a well-studied SDE in stochastic control [Dai Pra, 1991, Kappen et al., 2012, Tzen and Raginsky, 2019b, Fedus et al., 2018, Vargas et al., 2021a].

Prior work, such as aligned Schrodinger bridges [Somnath et al., 2023, Liu et al., 2023], have discussed this SDE in the context of dataset alignment and conditional generative modelling. Additionally, First Hitting Diffusion models [Ye et al., 2022] have explored a variant of PBM where instead of a Brownian motion Equation 9 is replaced with a VP-SDE. However, to our knowledge, PBM has not yet been compared carefully to VP-SDE within the context of generative modelling (some comparison has been done empirically for sampling [Vargas et al., 2023, Berner et al., 2022]).

## 3   EXPRESSIVENESS AND REGULARITY RESULTS

In this section, we present our main result. We demonstrate that $\nabla \ln \phi_t$ and thus the score of the OU-SDE can be approximated by a multi-layer neural network efficiently.

Theorem 3.1 in Tzen and Raginsky [2019b] provides neural network approximation and sampling guarantees for a different class of SDEs than DDPM (i.e. Equations 3 or 5). Thus in this section, we will adapt such results to denoising diffusion samplers [Vargas et al., 2023] and via directly relating the approximations to the score of the VP-SDE (Equation 2) we motivate how these results extend to DDPM based methods [Song et al., 2021b, Ho et al., 2020, Huang et al., 2021a].

Tzen and Raginsky [2019b] guarantee approximate sampling from a target distribution using a multilayer feedforward neural net drift, assuming the smoothness, Lipschitzness, and boundedness of $f(x) = \frac{\mathrm{d}\pi}{\mathrm{d}\mathcal{N}(0,\sigma^2 I)}(x)$, (Assumption B.2), as well as the smoothness of the activations (Assumption B.3) and uniform approximability of $f$ and its gradient by a neural network (Assumption B.4). In the following proposition and remark, we present our adaption of their results to DDS.

**Proposition 3.1.** *Suppose Assumptions in Appendix B are in force. Let $L$ denote the maximum of the Lipschitz constants of $f$ and $\nabla f$. Then for all $0 < \epsilon < 16L^2/c^2$, there exists*

a neural net $\hat{v} : \mathbb{R}^d \times [0,1] \to \mathbb{R}^d$ *with size polynomial in $1/\epsilon, d, L, c, 1/c$ such that the activation function of each neuron in the set of $\{\sigma, \sigma', ReLU\}$, and the following hold: If $\{\hat{x}_t\}_{t \in [0,1]}$ is the diffusion process governed by the Itô SDE:*

$$d\hat{x}_t = \hat{b}(\hat{x}_t, t)dt + \sqrt{2}dW_t, \qquad (12)$$

*with $x_0 \sim p_1 = \text{Law}(y_1) \approx \mathcal{N}(0, I)$ with the drift $\hat{b}(x,t) = -(x - 2\hat{v}(x, 1-t))$, then $\hat{\mu} := \text{Law}(\hat{x}_1)$, satisfies $D(\mu||\hat{\mu}) \leq \epsilon$.*

*Remark* 3.2. Assuming $\pi$ satisfies a logarithmic Sobolev inequality, extending the time domain to $t \in [0, T]$ and sampling $\hat{x}_0 \sim \mathcal{N}(0, I)$ approximately, it follows that $D(\mu||\hat{\mu}) \leq e^{-T}\text{KL}(\pi||\mathcal{N}(0,1)) + T\epsilon$.

The proof will closely follow Tzen and Raginsky [2019b] however key steps must be slightly modified to show that the value function satisfies the required regularity properties to exploit the core results in Tzen and Raginsky [2019a].

### 3.1 PRIOR WORK - HEAT SEMIGROUP AND FÖLLMER DRIFT

Here we will introduce the heat semigroup and Föllmer drifts to highlight the previous work done in [Tzen and Raginsky, 2019b].

**Definition 3.3.** The heat semi-group is defined as

$$Q_t^\sigma f(y) = \mathbb{E}_{Z \sim \mathcal{N}(0,I)} \left[ f\left(y + \sigma t^{1/2} Z\right) \right], \qquad (13)$$

and thus the Föllmer drift [Tzen and Raginsky, 2019b] can be expressed as

$$v_t^*(y) = \nabla \ln Q_{T-t}^\sigma f(y). \qquad (14)$$

Where $v^*$ can be used to sample exactly from the desired target distribution by simulating the Föllmer drift SDE in [Tzen and Raginsky, 2019b], which coincides exactly with the backwards pinned Brownian motion when setting $\alpha_t = \sigma^2 t$:

$$dy_t = \left\{ \overbrace{\frac{y_t}{t} + \sigma^2 \nabla \ln p_{T-t}(y_t)}^{v_t^*(y_t)} \right\} dt + \sigma dW_t, \ y_0 = 0.$$

This process is commonly referred to as the Schödinger Föllmer. The prior seminal work of Tzen and Raginsky [2019b] focuses on proving regularity properties of the heat semigroup as well as expressiveness remarks for the Föllmer drift. In this work, we port over these results to denoising diffusion models (i.e. VP-SDE based models).

### 3.2 OU SEMIGROUP AND TIME REVERSAL

This section introduces the OU semigroup [Metafune et al., 2002] whose logarithmic gradient can be directly connected

to the score [Song et al., 2021b] in Equation 2. Based on this reformulation of the score we can extend the results from Tzen and Raginsky [2019b] to denoising diffusion via VP-SDEs. In the remainder of this section, we will introduce new results pertaining to the regularity properties of this operator that will enable us to prove Proposition 3.1.

**Definition 3.4.** We define the VP-SDE semigroup as,

$$U_t^{\beta_t} f(y) = \mathbb{E}_{Z \sim \mathcal{N}(0,I)} \left[ f\left( e^{-\int_0^t \beta_s ds} y \right. \right.$$
$$\left. \left. + \sigma (1 - e^{-2\int_0^t \beta_s ds})^{1/2} Z \right) \right]. \qquad (15)$$

Then the OU-semigroup [Metafune et al., 2002] (typically defined with $\beta_t = \beta = 1$) is a simpler instance of the above

$$U_t^\beta f(y) = \mathbb{E}_{Z \sim \mathcal{N}(0,I)} \left[ f\left( e^{-\beta t} y + \sigma(1 - e^{-2\beta t})^{1/2} Z \right) \right].$$

For simplicity we will be working with the OU semi-group when $\beta = 1$ (denoted $U_t$), however, these results can be extended to the more general case. In the following remark, we highlight the connection between the OU semi-group, the value function, and the score in DDPM.

*Remark* 3.5. The drift of the time reversal of the VP-SDE (i.e. $b^*(y, t) = -\beta_{T-t}(y - 2\sigma^2 \nabla \ln \phi_{T-t}(y)))$ can be expressed in terms of the OU semigroup via:

$$\nabla \ln \phi_{T-t}(y) = \nabla_y \ln U_{T-t}^{\beta_t} f(y). \qquad (16)$$

When $f(x) = \frac{d\pi}{d\mathcal{N}(0,\sigma^2 I)}(x)$. This in turn can be related to the score

$$\nabla \ln p_{T-t}(y) = -\left( \frac{y}{2\sigma^2} - \nabla \ln \phi_{T-t}(y) \right)$$
$$= -\left( \frac{y}{2\sigma^2} - \nabla_y \ln U_{T-t}^{\beta_t} f(y) \right). \qquad (17)$$

From this stage on we consider the case where $\sigma = \beta = 1$. Notice how the formulation in Remark 3.5 is reminiscent of the Föllmer drift [Föllmer, 1984, Dai Pra, 1991, Tzen and Raginsky, 2019b, Huang et al., 2021b]. Finally, we highlight that it is this very simple remark that facilitates porting over the general proof strategy from [Tzen and Raginsky, 2019b] to diffusion-based models. Furthermore, we remind the reader that the results in Tzen and Raginsky [2019b] only apply to the Föllmer drift and the heat semigroup (i.e. $\nabla_y \ln \phi_t(y) = \nabla_y \ln Q_t f(y)$ with $Q_t f(y) = \mathbb{E}_{Z \sim \mathcal{N}(0,I)} \left[ f\left( y + \sqrt{t}Z \right) \right]$), thus requiring new results.

### 3.3 REGULARITY PROPERTIES

In this section, we will prove regularity properties pertaining to the OU semigroup which will allow us to extend the theoretical guarantees in Tzen and Raginsky [2019b] to denoising diffusion models and samplers [Song et al., 2021b, Ho et al., 2020, Vargas et al., 2023]. Moving forward we

prove a basic auxiliary result regarding the commutativity of the OU-semigroup with partial derivatives. From this result, by using Corollary E.1, we could bound the OU-semigroup norm when differentiated. Proofs for the following results can be found in Appendix C, and Appendix D.

**Lemma 3.6.** *OU semigroup is commutative with the gradient operator that is for $f : \mathbb{R}^d \to \mathbb{R}$ we have $\partial_{y_i} U_t f(y) = U_t \partial_{y_i} f(y)$.*

### 3.3.1 Terminal Cost

Contrary to Tzen and Raginsky [2019a] $g_{x,t}(z) = g(e^{-t}x + (1 - e^{-2t})^{1/2}z)$ (where $x \in B^d(R), z \in \mathbb{R}^d$)is not Lipschitz in a Euclidean sense. As a result the standard Euclidean covering-number properties used in Tzen and Raginsky [2019a] no longer apply and thus we have to derive bounds for these quantities from scratch.

Across this section will refer to $g_{x,t}(z)$ as the terminal cost, due to its role in stochastic control. We want to underline to the reader that this quantity is of high importance as the optimal drift can be expressed in terms of the OU-semigroup is applied to the terminal cost ($\nabla \ln \phi_t(x) = \nabla \ln U_t g_{x,t}(z)$) when $g = f$.

- First we prove that a centered version of the terminal cost is $\mathscr{L}^2(Q)$ Lipchitz with respect to a newly defined metric. This will allow us to obtain a bound for the covering number of a function class induced by the terminal cost.

- We then derive an envelope for the terminal cost. This in conjunction with further results on covering numbers allows us to control Dudley's entropy integral [Dudley, 1967]. This in turn enables results from empirical process theory [Giné and Nickl, 2021] that quantify the error for an empirical estimate of the OU semigroup.

**Lemma 3.7.** *($\mathscr{L}^2$ Lipchitz condition) Let $\bar{g}_{t,x}(z) = g(e^{-t}x + (1 - e^{-2t})^{1/2}z) - g(0)$ then it follows that:*

$$||\bar{g}_{t,x}(z) - \bar{g}_{t',x'}(z)||_{\mathscr{L}^2(Q)} \leq$$
$$L\left(1 + \sqrt{2}||z||_{\mathscr{L}^2(Q)}\right)\rho_{OU}((t,x),(t',x')),$$

*such that $\rho_{OU}((t,x),(t',x'))=||e^{-t}x - x'e^{-t'}|| + |t - t|^{1/2}$.*

**Lemma 3.8.** *Let $g : \mathbb{R}^d \to \mathbb{R}$ be L-Lipschitz with respect to the Euclidean norm. Then for $F(z) := L((R \vee 1) + \sqrt{2}||z||)$ we have:*

$$\left|g\left(e^{-t}x + (1 - e^{-2t})^{1/2}z\right) - g(0)\right| \leq F(z). \quad (18)$$

### 3.3.2 Covering Number

We must first obtain a notion of approximation "difficulty" in approximating the OU-semigroup to obtain estimation

errors on the score. To do so, we must obtain bounds quantifying how many "tiles" (closed balls) are required to cover the space $\mathcal{G}$. The formal mechanism to do so is known as a covering number.

The $\mathscr{L}^2(Q)$ covering number of the function space $\mathcal{G}$ is defined by:

$$N\left(\mathcal{G}, \mathscr{L}^2(Q), \varepsilon\right) := \min\left\{K : \exists f_1, \ldots, \exists f_K \in \mathscr{L}^2(Q)\right.$$
$$\left.\text{s.t.} \sup_{q \in \mathcal{G}} \min_{k \leq K} ||g - f_k||_{L^2(P)} \leq \varepsilon\right\}.$$

In general, the covering number $N(\mathcal{A}, \rho, \varepsilon)$ is the smallest number of balls of size $\epsilon$ wrt to the metric $\rho$ that cover the set $\mathcal{A}$. Once we obtain the appropriate bound on $N\left(\mathcal{G}, \mathscr{L}^2(Q), \varepsilon\right)$ the results from [Tzen and Raginsky, 2019b] follow with minor modifications and thus Corollary 3.1 will follow. In this section we will be bounding the $\mathscr{L}^2(Q)$ covering number of the function space $\mathcal{G} := \left\{\bar{g}_{x,t} : x \in \mathrm{B}^d(R), t \in [0, 1]\right\}$.

**Lemma 3.9.** *Given the metric space $\left([0, T] \times B^d(R), \rho_{OU}\right)$ where:*

$$\rho_{OU}((t,x),(t',x')) = ||e^{-t}x - x'e^{-t'}|| + |t - t'|^{1/2},$$

*and $||(t,x)||_{OU} = \rho_{OU}((t,x),(0,0)) = ||e^{-t}x|| + |t|^{1/2}$. It follows that:*

$$N(\mathcal{G}, \mathscr{L}^2(Q), \epsilon||F||_{\mathscr{L}^2(Q)}) \leq N([0,T] \times B^d(R), \rho_{OU}, \epsilon).$$

**Lemma 3.10.** *Given the metric space $\left([0, T] \times B^d(R), \rho_{OU}\right)$ it follows that:*

$$N([0,T] \times B^d(R), \rho_{OU}, \epsilon) \leq$$
$$N([0,T], |\cdot|, \epsilon^2/4)N(B^d(R), ||\cdot||, \epsilon/2). \quad (19)$$

From Lemmas 3.9, 3.10 it follows that:

$$N(\mathcal{G}, \mathscr{L}^2(Q), \epsilon||F||_{\mathscr{L}^2(Q)}) \leq$$
$$N([0,T], |\cdot|, \epsilon^2/4)N(B^d(R), ||\cdot||, \epsilon/2) \quad (20)$$

Establishing the existence of such bounds will facilitate our ability to subsequently demonstrate the approximation results. We will move forward in presenting two tighter bounds for the error. Proofs for Lemmas 3.9 and 3.10 can be found in Appendix D.

### 3.3.3 Sharper Bounds for OU Semigroup Covers

In this section, we will present two bounds concerning the metric space $\left([0, T] \times B^d(R), \rho_{OU}\right)$. The first bound is an extension of the heat semigroup results presented in Tzen and Raginsky [2019b] in $[0, 1]$ to $[0, T]$:

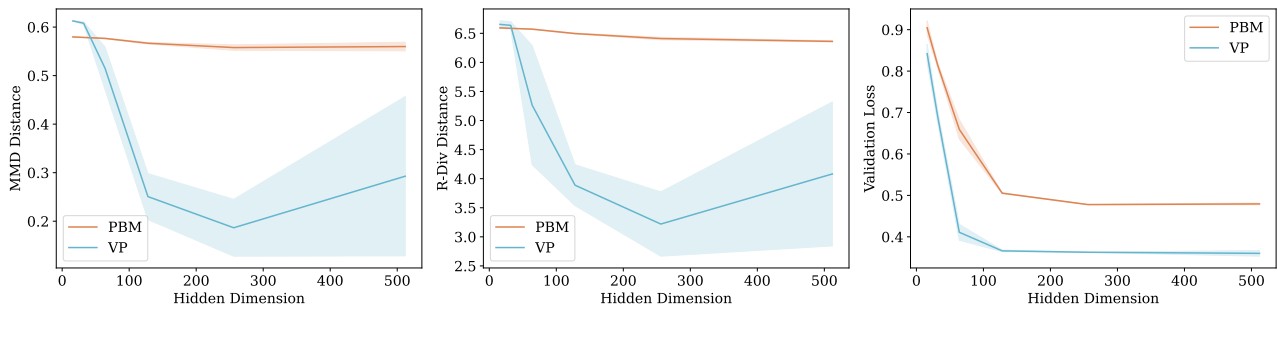

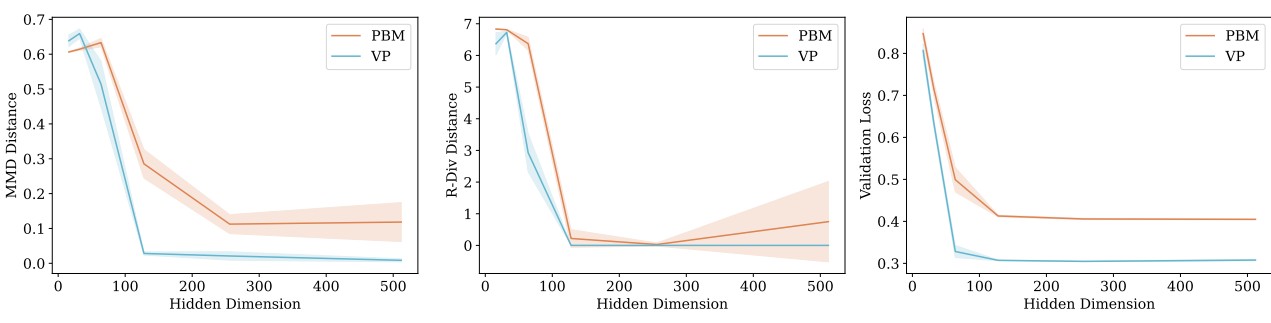

Figure 2: Comparison of distances between $\pi$ and $p_\theta^{\mathrm{model}}$ at time $T$ over 50-dimensional Funnel and GMM-10 datasets.

**Corollary 3.11.** *Given the metric space* $\left([0,T] \times B^d(R), \rho_{OU}\right)$ *it follows that:*

$$N(B^d(R), || \cdot ||, \epsilon/2)N([0,T], |\cdot|, \epsilon^2/4) \le \left(\frac{2\sqrt{3RT}}{\epsilon}\right)^{2d}.$$

The second bound is obtained from the properties of the OU process, establishing a tighter bound for this particular metric space.

**Proposition 3.12.** *Given the metric space* $\left([0,T] \times B^d(R), \rho_{OU}\right)$ *it follows that:*

$$N([0,T] \times B^d(R), \rho_{OU}, \epsilon) \le \left(\frac{2e^{-\epsilon^2/2}\sqrt{3TR}}{\epsilon}\right)^{d}.$$

### 3.4 SCORE ESTIMATION RESULTS

The following result constitutes an important piece in proving Proposition 3.1, since the way $N$ is picked depends on the previously mentioned Lemmas.

**Corollary 3.13.** *For any* $\varepsilon > 0$ *and any* $R > 0$*, there exist* $N = \mathrm{poly}(1/\varepsilon, d, L, R, T)$ *points* $z_1, \ldots, z_N \in \mathbb{R}^d$*, and for* $p(x,t,z_n) = e^{-t}x + (1-e^{-2t})^{1/2}z_n$*, for which the*

*following holds:*

$$\max_{n \le N} \|z_n\| \le 8\sqrt{(d+6)\ln N},$$

$$\sup_{x \in B^d(R)} \sup_{t \in [0,1]} \left| \sum_{n=1}^{N} \frac{\nabla f\left(p(x,t,z_n)\right)}{N} - U_t f(x) \right| \le \varepsilon,$$

$$\sup_{x \in B^d(R)} \sup_{t \in [0,1]} \left\| \sum_{n=1}^{N} \frac{\nabla f\left(p(x,t,z_n)\right)}{N} - \nabla U_t f(x) \right\| \le \varepsilon.$$

Following the steps from Tzen and Raginsky [2019b], Lemma 3.11 and Corollary 3.13, we arrive at one of the main results that will help prove Proposition 3.1.

Fortunately, the first bound derived in Lemma 3.11 arrives at a computable integral over the *Koltchinskii-Pollard $\epsilon$-entropy*, which is needed for the completion of the proof of Corollary 3.13, which then is used in proving Proposition 3.1 (see Appendix E). However, for the second bound, despite being tighter, the integral is not tractable. A more detailed description of this can be found in the Appendix proof of Lemma D.2, and the following remark.

Following results in Proposition 3.11 and Corollary 3.12 we can observe the presence of a $e^{-\frac{d\epsilon^2}{2}}$ term in the cover for the OU semi-group, thus its cover is smaller than that of the heat semigroup, this motivates the following observation:

**Observation 1.** *The Koltchinskii-Pollard $\epsilon$-entropy that cor-*

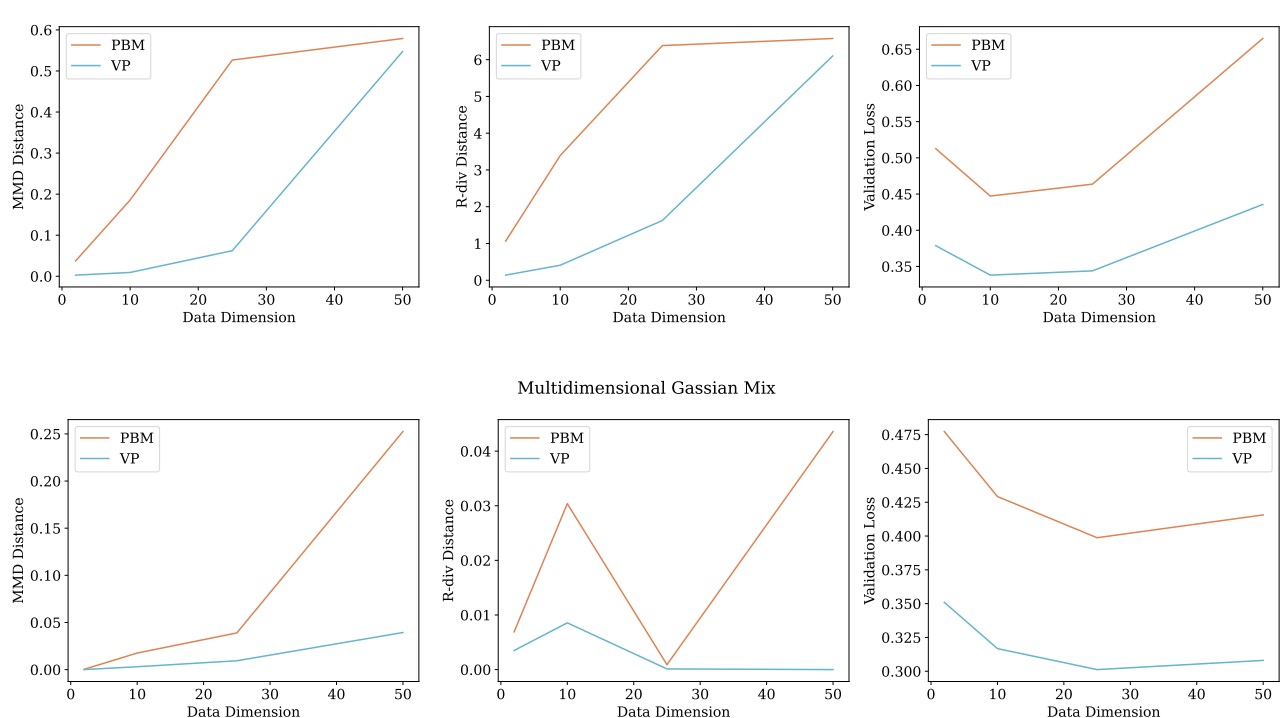

Figure 3: Comparison of distances between $\pi$ and $p_\theta^{\text{model}}$ at time $T$ over Funnel and Mixed Gaussians varying in dimensions.

*responds to the OU semigroup lower bounds that of the heat semi-group.*

We believe Observation 1 motivates how the score for VP-SDEs forms a simpler function class than the score for the PBM, which potentially indicates that the score on VP-SDE admits a neural network estimator achieving a smaller error than that of the score of a PBM [Tzen and Raginsky, 2019b], we will now explore this conjecture empirically.

# 4 SIMULATIONS

In this section, we evaluate the performance of VP-SDE and PBM using the score matching loss [Song et al., 2021b] together with other metrics across different network sizes and dataset dimensions. Section 4.1 analyzes performance on two synthetic datasets, and Section 4.2 compares the performance on image data. Detailed information on learning rates, dataset splits, training epochs, and noise schedules is provided in Appendices G and H, respectively.

## 4.1 SYNTETHIC SIMULATIONS

We explore both VP-SDE and PBM across two simulated datasets, selected due to their flexibility in being able to increase the dimension of space.

**GMM-10:** We use a Gaussian mixture model with 10 mixtures, each mixture is parameterized $\mathcal{N}(\mu_i, I)$ where $\mu_i$ is sampled uniformly within the $(d-1)$-ball of Radius 6.

**Neals Funnel [Neal, 2011]** This d-dimensional challenging distribution is given by $\gamma(x_{1:d}) = \mathcal{N}(x_1; 0, \sigma_f^2)\mathcal{N}(x_{2:d}; 0, \exp(x_1)I)$, where $\sigma_f^2 = 9$.

### 4.1.1 Evaluation Metrics

To assess which method we report the following performance metrics across a series of numerical simulations:

**Score matching loss**: We report the score matching loss [Song et al., 2021b] on a hold test set. This loss acts as a proxy to measure how well the trained network has learned the score.

**MMD**: We the use maximum mean discrepancy metric [Gretton et al., 2012] to measure the distance between $\mathcal{D}(p_\theta^{\text{model}}, \pi)$. The motivation for this is that the KL-divergence between the marginals $p_\theta^{\text{model}}$ and $\pi$ (via data processing and Girsanov Theorem) is upper bounded by the error between the scoring network and the true score, thus a better performance in score matching typically indicates better marginal performance, here we assess the latter.

**r-divergence**: Similar to the MMD experiments we explore an additional divergence (the r-divergence [Zhao and Cao,

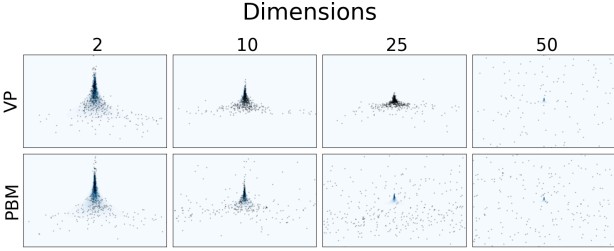

Dimensions

Figure 4: Samples (2D slice $(x_1, x_0)$) from PBM and VP trained on various sizes of the Funnel distribution. In the background probability density of the Funnel distribution.

2023]) to more thoroughly verify the performance in sampling the data distribution.

Note for the MMD and r-divergence metrics we use $1000$ samples from the trained score models and the target distributions to compute the aforementioned metrics and $20000$ samples for the validation score matching loss. More details can be found in Appendix G.

#### 4.1.2 Score Estimation Across Network Widths

For this experiment, we fix the dimensions of the data sets to $d = 50$ and vary the width of the score networks across $4, 16, 32, 64, 128, 256, 512$. From our results, we can see that in Figure 2 on the left-hand side, VP-SDE attains a lower score-matching loss for the same number of parameters and can sample the target distribution better than PBM, suggesting that VP-SDE requires less expressive networks to be estimated, which is in agreement with the insights we obtained from our covering number results.

In Figure 2, for the Funnel dataset, VP seems to express a double descent Nakkiran et al. [2021], d'Ascoli et al. [2020] type of behavior. As the hidden layer dimension passes $256$ parameters, the model generates samples that are further away from $\pi$.

We also ran experiments for $d = 10$, and the same behavior can be noticed across all network widths. The experiments can be viewed in Appendix G.2. In this case, for the Funnel dataset, the double descent behavior can be noticed in both VP and PBM cases, for MMD and R-Div metrics.

#### 4.1.3 Score Estimation Across Data-set Dimensions

In this sequence of experiments depicted in Figure 3 on the right, we maintain a fixed network width of $64$ while varying the dimensionality across $2, 10, 25, 50$ to evaluate the performance of both methods in estimating the score as the dimensionality of the target samples increases. For dimensions below $50$, VP tends to sample points that match the target/data distribution more closely on the Funnel dataset, whereas, at $50$ dimensions, both their performances start

to degrade similarly (e.g. also see Figure 4). Finally, VP consistently produces superior samples in the case of the GMM targets, thus corroborating our observation.

### 4.2 IMAGE DATA SIMULTATION

In this section, we investigate the performance of VP-SDE and PBM on image datasets, specifically MNIST, MNIST-Fashion, and CIFAR-10. We conducted all experiments using a simple single-block ResNet architecture with three feature maps with a total number of $1.34$ million parameters (only $3\%$ of the number of the parameters used in Song et al. [2020]). Given the nature of image datasets, we evaluated the quality of generated samples using the Frechet Inception Distance (FID) Jiralerspong et al. [2023]. The nature of the experiment is to see the difference in the relative performance of the two forward processes. Further details, and generated samples are provided in Appendix H.

Table 1: Comparison of FID scores on the test set of MNIST, Fashion MNIST, and CIFAR-10 datasets

| Dataset | MNIST | Fashion MNIST | CIFAR-10 |
|---|---|---|---|
| VP-SDE | $13.86 \pm 0.06$ | $6.99 \pm 0.03$ | $40.09 \pm 0.88$ |
| PBM | $15.59 \pm 0.07$ | $17.26 \pm 0.35$ | $49.47 \pm 0.67$ |

As shown in Table 1, given the same computation/parameter budget, VP achieves better estimates than PBM across all image datasets, with a notable improvement in performance observed in MNIST-Fashion. This suggests that the balance between the complexity of the datasets and the size of the network was optimized for this particular dataset.

## 5 CONCLUSION

We establish a connection between the VP-SDE score and the OU-semigroup, revealing similarities between Föllmer drift-based and DDPM-based sampling approaches. Using this connection, we demonstrate how the VP-SDE score can be approximated efficiently by multilayer neural networks, under fairly general assumptions on the target distribution. To exploit previous results on the Föllmer drift [Tzen and Raginsky, 2019b] we establish novel regularity properties for the OU-semigroup that allow us to adapt the results in Tzen and Raginsky [2019b] to our setting. Finally motivated by our theoretical results we empirically demonstrate how a VP-SDE based forward process can be approximated better by a neural network of the same size than one with a PBM-SDE forward process.

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

# A  LIST OF DETAILED CONTRIBUTIONS

Main contributions:

1. Firstly, Vargas et al. [2023] prompts us to consider Remark 3.5, which allows us to view time reversal in denoising diffusions through the lens of stochastic control. Unfortunately, this did not enable us to directly apply results from Tzen and Raginsky [2019b]. The regularity properties for the OU semi-group and the Heat semi-group are significantly different, thus rendering their entropy integral bounds inapplicable.

2. The quantity $\bar{g}_{t,x}(z)$ for the heat semi-group is Lipschitz relative to the Euclidean metric. However, this is not the case for OU. Instead, we can obtain a Lipschitz property with respect to a different metric, results presented in Section 3.3.1:

$$\rho_{OU}\left((t,x),(t',x')\right) = ||e^{-t}x - x'e^{-t'}|| + |t - t'|^{1/2}$$

3. Once we have established a Lipschitz property, the next step is to obtain a covering number for the space of functions induced by $\bar{g}_{t,x}(z)$ (indexed by $x$ and $t$). Unfortunately, as the Lipschitz property is not Euclidean, we can no longer use the simple covering lemma employed in Tzen and Raginsky [2019b]:

$$N\left(\mathcal{G}, L^2(Q), \varepsilon\|F\|_{L^2(Q)}\right) \leq N\left(\mathrm{B}^d(R), \|\cdot\|, \varepsilon/2\right) \cdot N\left([0,1], |\cdot|, \varepsilon^2/4\right)$$

This standard result known for covers of function spaces with a Euclidean Lipschitz property Shalev-Shwartz and Ben-David [2014] does not apply to our metric space.

4. We then construct the cover for $([0,T] \times B^d(R), \rho_{OU})$ ourselves, which required several non-trivial steps (see the proof for Lemma 3.10). A significant portion of our work was devoted to this challenging step (the whole process is described in Section 3.3.2):

$$N\left([0,T] \times B^d(R), \rho_{OU}, \epsilon\right) \leq N\left([0,T], |\cdot|, \epsilon^2/4\right) N\left(B^d(R), \|\cdot\|, \epsilon/2\right)$$

5. Once we obtained step 4, the derived bounds were slightly different, requiring some symbol manipulation and further simplification steps to achieve the final results. In summary, step 4 was a challenging and distinguishing aspect of our work compared to Tzen and Raginsky [2019b]. Furthermore, to the best of our knowledge, our work is the first to derive these mathematical properties of the OU semi-group.

6. Deriving two bounds for OU semi-group in Corollary 3.11 and Proposition 3.12. Since the bound derived in Proposition 3.12 is tighter we arrived at Observation 1, which suggests that for the same parameterization, and target distribution, VP-SDE-based models should perform better than PBM ones.

7. We derive the transition Kernel for the PBM-SDE which is a generalisation of the Föllmer drift explored in [Tzen and Raginsky, 2019b] (Section 2.4, Appendix F for full derivations), this allowed us to empirically compare PBM-SDE based score matching to VP-SDE in order to see which scales better when limited to the same neural network expressiveness.

8. Empirically tested Observation 1 on two datasets of different sizes, as well as on models with varying width dimensions, in Section 4.

# B  ASSUMPTIONS

**Assumption B.1.** Throughout all this work we assume that the target distribution $\pi$ has a density that is it is absolutely continuous wrt to the Lebesgue measure on $\mathbb{R}^d$.

**Assumption B.2.** The function $f$ is differentiable, both $f$ and $\nabla f$ are L-Lipschitz, and there exists a constant $c \in (0,1]$ such that $f \geq c$ everywhere.

**Assumption B.3.** The activation function $\sigma : \mathbb{R} \to \mathbb{R}$ is differentiable. Moreover, there exists $c_\sigma > 0$ depending only on $\sigma$, such that the following holds: For any L-Lipschitz function $h : \mathbb{R} \to \mathbb{R}$ which is constant outside the interval $[-R, R]$ and for any $\delta > 0$, there exist real numbers $a, \{\alpha_i, \beta_i, \gamma_i\}_{i=1}^m$ where $m \leq c_\sigma \frac{RL}{\delta}$, such that the function $\tilde{h}(x) = a + \sum \alpha_i \sigma(\beta_i x + \gamma_i)$ satisfies $\sup_{x \in \mathbb{R}} |\tilde{h}(x) - h(x)| \leq \delta$.

Finally as per Tzen and Raginsky [2019b] we introduce the assumption pertaining to the approximability of $f$ by neural nets. Let $\sigma : \mathbb{R} \to \mathbb{R}$ be a fixed nonlinearity. Given a vector $w \in \mathbb{R}^n$ and scalars $\alpha, \beta$, define the function

$$N^\sigma_{w,\alpha,\beta} : \mathbb{R}^n \to \mathbb{R}, \quad N^\sigma_{w,\alpha,\beta}(x) := \alpha \cdot \sigma\left(w^T x + \beta\right).$$

For $\ell \geq 2$, we define the class $\mathcal{N}^\sigma_\ell$ of $\ell$-layer feedforward neural nets with activation function $\sigma$ recursively as follows: $\mathcal{N}^\sigma_2$ consists of all functions of the form $x \mapsto \sum_{i=1}^m N^\sigma_{w_i,\alpha_i,\beta_i}(x)$ for all $m \in \mathbb{N}, w_1, \ldots, w_m \in \mathbb{R}^d, \alpha_1, \ldots, \alpha_m, \beta_1, \ldots, \beta_m \in \mathbb{R}$, and, for each $\ell \geq 2$,

$$\mathcal{N}^\sigma_{\ell+1} := \bigcup_{k \geq 1} \bigcup_{m \geq 1} \left\{ x \mapsto \sum_{i=1}^m N^\sigma_{w_i,\alpha_i,\beta_i}\left(h_1(x), \ldots, h_k(x)\right) : \right.$$
$$\left. \alpha_1, \ldots, \alpha_m, \beta_1, \ldots, \beta_m \in \mathbb{R}, w_1, \ldots, w_m \in \mathbb{R}^k, h_1, \ldots, h_k \in \mathcal{N}^\sigma_\ell \right\}.$$

**Assumption B.4.** For any $R > 0$ and $\epsilon > 0$, there exist a neural net $\hat{f} \in \mathcal{N}^\sigma_{l,s}$ with $l, s < \text{poly}(1/\epsilon, d, L, R)$, such that

$$\sup_{x \in B^d(R)} |f(x) - \hat{f}(x)| \leq \epsilon \quad and \quad \sup_{x \in B^d(R)} \|\nabla f(x) - \nabla \hat{f}(x)\| \leq \epsilon. \tag{21}$$

# C  REGULARITY RESULTS

**Remark 3.5.** *The time reversal of the VP-SDE (i.e. $b^*(y,t) = -\beta_{T-t}(y - 2\sigma^2 \nabla \ln \phi_{T-t}(y)))$ can be expressed in terms of the OU semigroup via:*

$$\nabla \ln \phi_{T-t}(y) = \nabla_y \ln U^{\beta_t}_{T-t} f(y), \tag{22}$$

*When $f(x) = \frac{\pi}{\mathcal{N}(0,\sigma^2 I)}(x)$. This in turn can be related to the score*

$$\nabla \ln p_{T-t}(y) = -\left(\frac{y}{2\sigma^2} - \nabla \ln \phi_{T-t}(y)\right) = -\left(\frac{y}{2\sigma^2} - \nabla_y \ln U^{\beta_t}_{T-t} f(y)\right). \tag{23}$$

*Proof.* Consider the OU semigroup evaluated on the appropriate RND:

$$U^{\beta_t}_t f(y) = \mathbb{E}_{Z \sim \mathcal{N}(0,I)}\left[\frac{\pi}{\mathcal{N}(0,\sigma^2 I)}\left(e^{-\beta t} y + \sigma(1 - e^{-2\beta t})^{1/2} Z\right)\right]$$

$$= \mathbb{E}_{x_T \sim p^{\text{ref}}_{T|t}(\cdot|x)}\left[\frac{\pi}{\mathcal{N}(0,\sigma^2 I)}(x_T)\right]$$

$$= \int p^{\text{ref}}_{T|t}(x_T|x) \frac{\pi}{\mathcal{N}(0,\sigma^2 I)}(x_T)\, dx_T$$

$$= \int \frac{p^{\text{ref}}_{t|T}(x|x_T) p^{\text{ref}}_T(x_T)}{p^{\text{ref}}_t(x)} \frac{\pi}{\mathcal{N}(0,\sigma^2 I)}(x_T)\, dx_T$$

$$= \int \frac{p^{\text{ref}}_{t|T}(x|x_T)}{p^{\text{ref}}_t(x)} \pi(x_T)\, dx_T = \frac{p_t(x)}{p^{\text{ref}}_t(x)}$$

and thus it follows that

$$\nabla \ln p_{T-t}(y) = -\left(\frac{y}{2\sigma^2} - \nabla_y \ln U^{\beta_t}_{T-t} f(y)\right). \tag{24}$$

relating the score and the OU semi-group as required. $\square$

**Lemma 3.6.** *OU semigroup is commutative with the gradient operator that is for $f : \mathbb{R}^d \to \mathbb{R}$ we have $\partial_{y_i} U_t f(y) = U_t \partial_{y_i} f(y)$.*

*Proof.* It suffices to show that

$$d(x,z) = \delta^{-1}(f(e^{-t} x + (1 - e^{-2t})^{1/2} z) - f(e^{-t}(x + \delta e_i) + (1 - e^{-2t})^{1/2} z)), \tag{25}$$

is dominated, where $[e_i]_j = \delta_{ij}$. As $f$ is Lipchitz by assumption it follows that

$$|d(x,z)| \leq L|\delta^{-1}e^{-t}\delta| = Le^{-t} \leq L \tag{26}$$

As $L$ is integrable under $\mathcal{N}(0, I)$ we have shown $d(x, z)$ is dominated for all $\delta$ and thus the partial derivative operator and the OU semigroup commute. $\square$

The choice of $F(z) := L((R \vee 1) + \sqrt{2}||z||)$ with these specific constants arises from the following result.

**Lemma 3.7.** ($\mathcal{L}^2$ *Lipchitz condition*) *Let* $\bar{g}_{t,x}(z) = g(e^{-t}x + (1 - e^{-2t})^{1/2}z) - g(0)$ *then it follows that:*

$$||\bar{g}_{t,x}(z) - \bar{g}_{t',x'}(z)||_{\mathcal{L}^2(Q)} \leq L\left(1 + \sqrt{2}||z||_{\mathcal{L}^2(Q)}\right)\rho_{OU}((t,x),(t',x'))$$

*such that:*

$$\rho_{OU}((t,x),(t',x')) = ||e^t x - x'e^{t'}|| + |t - t'|^{1/2} \tag{27}$$

*Proof.*

$$\begin{aligned}
||\bar{g}_{t,x}(z) - \bar{g}_{t',x'}(z)||_{\mathcal{L}^2(Q)} &\leq L\big|\big|||e^{-t}x + (1 - e^{-2t})^{1/2}z - e^{-t'}x' - (1 - e^{-2t'})^{1/2}z|||\big|\big|_{\mathcal{L}^2(Q)} \\
&\leq L\Big|\Big|||e^{-t}x - e^{-t'}x'|| + |(1 - e^{-2t})^{1/2} - (1 - e^{-2t'})^{1/2}| \cdot ||z||\Big|\Big|_{\mathcal{L}^2(Q)} \\
&\leq L\left(||e^{-t}x - e^{-t'}x'|| + |(1 - e^{-2t})^{1/2} - (1 - e^{-2t'})^{1/2}| \cdot ||z||_{\mathcal{L}^2(Q)}\right) \\
&\leq L\left(||e^{-t}x - e^{-t'}x'|| + |e^{-2t} - e^{-2t'}|^{1/2} \cdot ||z||_{\mathcal{L}^2(Q)}\right) \\
&\leq L\left(||e^{-t}x - e^{-t'}x'|| + \sqrt{2}|t - t'|^{1/2} \cdot ||z||_{\mathcal{L}^2(Q)}\right)
\end{aligned}$$

Where in the last line we use that $\sup_{t \in [0,T]} |(e^{-2t})'| = 2$ and thus $e^{-2t}$ is 2-Lipchitz. $\square$

**Lemma 3.8.** *Let* $g : R^d \to R$ *to L-Lipschitz with respect to the Euclidean norm. Then for* $F(z) := L((R \vee 1) + \sqrt{2}||z||)$.

$$\left|g\left(e^{-t}x + (1 - e^{-2t})^{1/2}z\right) - g(0)\right| \leq F(z) \tag{28}$$

*Proof.* By Lipschitz continuity for all $z \in R^d, X \in B^d(R), t \in [0, T]$ we have:

$$|g\left(e^{-t}x + (1 - e^{-2t})^{1/2}z\right) - g(0)| \leq L||e^{-t}x + (1 - e^{-2t})^{1/2}z|| \tag{29}$$

$$\leq L(e^{-t}||x|| + (1 - e^{-2t})^{1/2}||z||) \tag{30}$$

Since both $e^{-t}$ and $(1 - e^{-2t})^{1/2}$ are strictly smaller than 1, we have:

$$L(e^{-t}||x|| + (1 - e^{-2t})^{1/2}||z||) \leq L(R + ||z||) \tag{31}$$

$$\leq L((R \vee 1) + ||z||) \leq F(z) \tag{32}$$

$\square$

# D COVERING NUMBER RESULTS

*Remark* D.1. The space $\left([0,T] \times B^d(R), \rho_{OU}\right)$ is a metric space, where

$$\rho_{OU}((t,x),(t',x')) = ||e^{-t}x - x'e^{-t'}|| + |t-t'|^{1/2}. \tag{33}$$

*Proof.* • **Positive definiteness**:

$$\rho_{OU}((t,x),(t',x')) = 0 \iff \tag{34}$$
$$||e^{-t}x - x'e^{-t'}|| + |t-t'|^{1/2} = 0 \iff \tag{35}$$
$$x = x' \text{ and } t = t. \tag{36}$$

Since in (35) both terms are positive on the LHS, each has to be $0$ to get the RHS, thus we get (36).

• **Symmetry**:

$$\rho_{OU}((t,x),(t',x')) = \rho_{OU}((t',x'),(t,x)). \tag{37}$$

• **Triangle inequality**: we show triangle inequality on $(t,x), (t',x')$ and $(t'',x'')$. First let us note, that $||e^{-t}x - x'e^{-t'}|| + ||e^{-t'}x' - x''e^{-t''}|| \geq ||e^{-t}x - x''e^{-t''}||$, since $||\cdot||$ has the triangle inequality. Now:

$$|t-t'|^{1/2} + |t'-t''|^{1/2} \geq |t-t''|^{1/2} \iff \tag{38}$$
$$|t-t'| + 2|t-t'|^{1/2}|t'-t''|^{1/2} + |t'-t''| \geq |t-t''|. \tag{39}$$

(39) is true, since $|\cdot|$ has the triangle inequality and $2|t-t'|^{1/2}|t'-t''|^{1/2} \geq 0$.

$\square$

**Lemma 3.9.** *Given the metric space $\left([0,T] \times B^d(R), \rho_{OU}\right)$ where:*

$$\rho_{OU}((t,x),(t',x')) = ||e^{-t}x - x'e^{-t'}|| + |t-t'|^{1/2} \tag{40}$$

*and*

$$||(t,x)||_{OU} = \rho_{OU}((t,x),(0,0)) = ||e^{-t}x|| + |t|^{1/2} \tag{41}$$

*It follows that:*

$$N(\mathcal{G}, \mathscr{L}^2(Q), \epsilon||F||_{\mathscr{L}^2(Q)}) \leq N([0,T] \times B^d(R), \rho_{OU}, \epsilon) \tag{42}$$

*Proof.* Consider the $\epsilon$-cover $A_{\rho_{OU}}$ with respect to $\rho_{OU}$ of $[0,T] \times B^d(R)$ it follows that for any $(t,x) \in [0,T] \times B^d(R)$ we have that there exists $(t',x') \in A_{\rho_{OU}}$ such that $\rho_{OU}((t,x),(t',x')) \leq \epsilon$ then by Lemma 3.7 it follows that

$$||\bar{g}_{t,x}(z) - \bar{g}_{t',x'}(z)||_{\mathscr{L}^2(Q)} \leq L\left(1 + \sqrt{2}||z||_{\mathscr{L}^2(Q)}\right)\rho_{OU}((t,x),(t',x')) \tag{43}$$
$$\leq ||F||_{\mathscr{L}^2(Q)}\rho_{OU}((t,x),(t',x')) \tag{44}$$
$$\leq ||F||_{\mathscr{L}^2(Q)}\epsilon \tag{45}$$

Hence the set:

$$\mathcal{G}_{\rho_{OU}} = \{\bar{g}_{t,x} : (t,x) \in A_{\rho_{OU}}\} \tag{46}$$

is an $||F||\epsilon$ cover of $\mathcal{G}$ with respect to the metric $\rho_{OU}$

$\square$

**Lemma 3.10.** *We have that*

$$N([0,T] \times B^d(R), \rho_{OU}, \epsilon) \leq N([0,T], |\cdot|, \epsilon^2/4)N(B^d(R), ||\cdot||, \epsilon/2) \tag{47}$$

*Proof.* Let $B_{r_0}^d(R)$ denote a euclidean d-dimensional ball of radius $R$ centered at $r_0$ and let $B_{t_0\oplus x_0,\rho}^{d+1}(R')^2$ denote it's counterpart with respect to the metric $\rho$. Now notice that if $||e^{-t}x - e^{-t_0}x_0|| + |t - t_0|^{1/2} \leq \epsilon$ then $||e^{-t_0}(x - x_0)|| \leq \epsilon$ and $||x - x_0|| \leq e^{t_0}\epsilon$, thus,

$$\{t_0\} \times B_{x_0}^d(\epsilon) \subseteq \{t_0\} \times B_{x_0}^d(e^{t_0}\epsilon) \subseteq B_{t_0\oplus x_0,\rho}^{d+1}(\epsilon), \tag{48}$$

then since $\{t_0\} \times B_{x_0}^d(e^{t_0}\epsilon) \subseteq B_{t_0\oplus x_0,\rho}^{d+1}(\epsilon)$ we can construct an $\epsilon$ cover namely $A_{t_0}$ of $\{t_0\} \times B^d(R)$ with $N(B^d(R), ||\cdot||, \epsilon e^{t_0})$ balls. Finally notice that if $||e^{-t}x - e^{-t_0}x_0|| + |t - t_0|^{1/2} \leq \epsilon$ it follows that $|t - t_0|^{1/2} \leq \epsilon$ thus $[0, T]$ can be covered in $N([0, T], |\cdot|, \epsilon^2) \leq T\epsilon^{-2}$ sub intervals.

Let $U_T$ be the smallest cover containing $N([0, T], |\cdot|, \epsilon^2)$ intervals $u_n$ each centered at $t_n$, then:

$$A = \bigcup_{u_n \in U_T} A_{t_n} \tag{49}$$

is an $\epsilon$ cover of $[0, T] \times B^d(R)$ (with respect to the metric $\rho_{OU}$), notice this follows as $\forall x \in B^d(R)$ there exists an $x_0$ such that

$$[t_n - \epsilon^2, t_n + \epsilon^2] \times \{x\} \subseteq B_{t_n\oplus x_0,\rho}^{d+1}(\epsilon) \in A_{t_n} \tag{50}$$

Now we can see that

$$|A| \leq |U_T||A_0| = N([0, T], |\cdot|, \epsilon^2)N(B^d(R), ||\cdot||, \epsilon e^{t_0}), \tag{51}$$
$$\leq N([0, T], |\cdot|, \epsilon^2/4)N(B^d(R), ||\cdot||, \epsilon/2), \tag{52}$$

where $|A_0| = \max_n |A_{t_n}|$, completing our proof.

$\square$

**Lemma 3.11.** *Given the metric space* $([0, T] \times B^d(R), \rho_{OU})$ *it follows that:*

$$N(B^d(R), ||\cdot||, \epsilon/2)N([0, T], |\cdot|, \epsilon^2/4) \leq \left(\frac{2\sqrt{3RT}}{\epsilon}\right)^{2d} \tag{53}$$

*Proof.* We know the covering number of $N(B^d(R), ||\cdot||, \epsilon)$ is $\left(\frac{2R}{\epsilon}\right)^d$, and for $N([0, T], |\cdot|, \epsilon)$, it is $\frac{T}{2\epsilon}$. In our settings:

$$N(B^d(R), ||\cdot||, \epsilon/2)N([0, T], |\cdot|, \epsilon^2/4) \leq \left(\frac{6R}{\epsilon}\right)^d \left(\frac{2T}{\epsilon^2}\right) \tag{54}$$

Now for $d \geq 2$ and $\epsilon$ small enough ($\epsilon^{d-2} \leq (2T)^{d-1}$) we get $\left(\frac{2T}{\epsilon^2}\right) \leq \left(\frac{2T}{\epsilon}\right)^d$ After this, our inequality will become:

$$\left(\frac{6R}{\epsilon}\right)^d \left(\frac{2T}{\epsilon^2}\right) \leq \left(\frac{6R}{\epsilon}\right)^d \left(\frac{2T}{\epsilon}\right)^d = \left(\frac{2\sqrt{3RT}}{\epsilon}\right)^{2d}$$

$\square$

**Proposition 3.12.** *Given the metric space* $([0, T] \times B^d(R), \rho_{OU})$ *it follows that:*

$$N([0, T] \times B^d(R), \rho_{OU}, \epsilon) \leq \left(\frac{2e^{-\epsilon^2/2}\sqrt{3TR}}{\epsilon}\right)^d \tag{55}$$

---

$^2 a \oplus b$ denotes the concatenation of $a$ and $b$.

*Proof.* Let $\rho_{OU} = \rho$ and $B_{r_0}^d(R)$ denote a euclidean d-dimensional ball of radius $R$ centered at $r_0$ and let $B_{t_0 \oplus x_0, \rho}^{d+1}(R')$ denote it's counterpart with respect to the metric $\rho$. Now notice that if $||e^{-t}x - e^{-t_0}x_0|| + |t - t_0|^{1/2} \leq \epsilon$ then $||e^{-t_0}(x - x_0)|| \leq \epsilon$ and $||x - x_0|| \leq e^{t_0}\epsilon$, thus,

$$\{t_0\} \times B_{x_0}^d(\epsilon) \subseteq \{t_0\} \times B_{x_0}^d(e^{t_0}\epsilon) \subseteq B_{t_0 \oplus x_0, \rho}^{d+1}(\epsilon), \tag{56}$$

then since $\{t_0\} \times B_{x_0}^d(e^t\epsilon) \subseteq B_{t \oplus x_0, \rho}^{d+1}(\epsilon)$ we can construct an $\epsilon$ cover namely $A_{t_0}$ of $\{t_0\} \times B^d(R)$ with $\left(6R\epsilon^{-1}e^{-t_0}\right)^d$ balls of form the form $B_{t_0 \oplus x_0, \rho}^{d+1}$. Finally notice that if $||e^{-t}x - e^{-t_0}x_0|| + |t - t_0|^{1/2} \leq \epsilon$ it follows that $|t - t_0|^{1/2} \leq \epsilon$ thus $[0, T]$ can be covered in $2^{-1}T\epsilon^{-2}$.

picking the cover $U_T$ such that its elements $u_n$ are centered at $(n+1)\epsilon^2/2$ , then:

$$A = \bigcup_{u_n \in U_T} A_{(n+1)\epsilon^2/2} \tag{57}$$

is an $\epsilon$ cover of $[0, T] \times B^d(R)$ (with respect to the metric $\rho_{OU}$), notice this follows as $\forall x \in B^d(R)$ there exists an $x_0$ such that

$$[(n+1)\epsilon^2/2 - \epsilon^2, (n+1)\epsilon^2/2 + \epsilon^2] \times \{x\} \subseteq B_{(n+1)\epsilon^2/2 \oplus x_0, \rho}^{d+1}(\epsilon), \tag{58}$$

with $B_{(n+1)\epsilon^2/2 \oplus x_0, \rho}^{d+1} \in A_{(n+1)\epsilon^2/2}$.

Now we can see that $|A| \leq |U_T||A_0|$ ( $|A_0| = \max_n |A_{(n+1)\epsilon^2/2}|$) completing our proof. □

From Lemmas 3.9, 3.10 it follows that :

$$N(\mathcal{G}, \mathscr{L}^2(Q), \epsilon||F||_{\mathscr{L}^2(Q)}) \leq N([0, T], |\cdot|, \epsilon^2/4)N(B^d(R), ||\cdot||, \epsilon/2) \tag{59}$$

**Lemma D.2.** *The Koltchinskii-Pollard $\varepsilon$-entropy of $N(\mathcal{G}, \mathscr{L}^2(Q), \epsilon||F||_{\mathscr{L}^2(Q)})$ is given by*

$$H(\mathcal{G}, F, \varepsilon) := \sup_Q \sqrt{\ln 2N\left(\mathcal{G}, L^2(Q), \varepsilon||F||_{L^2(Q)}\right)}$$

*Then we have*

$$J(\mathcal{G}, \mathscr{L}^2(Q)) = \int_0^\infty H(\mathcal{G}, F, \varepsilon)\mathrm{d}\varepsilon \leq 2\sqrt{3\pi RdT}$$

*with $H(\mathcal{G}, F, \varepsilon) \leq \sqrt{\left(4d\ln\frac{2\sqrt{3RT}}{\varepsilon}\right)_+}$.*

*Proof.* Following the derivations from Tzen and Raginsky [2019b], and our bound from Lemma 3.11:

$$J(\mathcal{G}, \mathscr{L}^2(Q)) = \int_0^\infty H(\mathcal{G}, F, \varepsilon)\mathrm{d}\varepsilon \leq \int_0^\infty \sqrt{\left(4d\ln\frac{2\sqrt{3RT}}{\varepsilon}\right)_+}d\epsilon \tag{60}$$

$$= 2\sqrt{d}\int_0^{2\sqrt{3R}} \sqrt{\left(\ln\frac{2\sqrt{3RT}}{\varepsilon}\right)}d\epsilon = \tag{61}$$

$$= 4\sqrt{3dRT}(ye^{-y^2}\Big|_\infty^0 - \int_\infty^0 e^{-y^2}) = 4\sqrt{3dRT}\frac{\sqrt{\pi}}{2} = 2\sqrt{3dR\pi T} \tag{62}$$

□

Thus by Lemma D.4 (see the start of Page 18 in Tzen and Raginsky [2019a]) we now have the following corollary of Lemma C.4 from Tzen and Raginsky [2019b]

**Corollary D.3.** *(Theorem C.4. from Tzen and Raginsky [2019b]) Let $g : \mathbb{R}^d \to \mathbb{R}$ be L-Lipschitz with respect to the Euclidean norm. Let $Z_1, \ldots, Z_N$ be i.i.d. copies of a d-dimensional random vector $Z$, such that $U := \|Z\|$ has finite $\psi_2$ norm. Then there exists an absolute constant $C > 0$, such that, for any $\gamma > 0$,*

$$\sup_{x \in \mathrm{B}^d(R)} \sup_{t \in [0,1]} \left| \frac{1}{N} \sum_{n=1}^{N} g\left(e^{-t}x + (1 - e^{-2t})^{1/2}Z_n\right) - \mathbb{E}\left[g\left(e^{-t}x + (1 - e^{-2t})^{1/2}Z\right)\right] \right|$$

$$\leq C \left[ \frac{16L\sqrt{6\pi Rd}\left((R \vee 1) + \|U\|_{\psi_2}\right)}{\sqrt{N}} + 5L\left((R \vee 1) + \|U\|_{\psi_2}\right)\sqrt{\frac{\gamma}{N}} \right]$$

*with probability at least $1 - e^{-\gamma}$.*

Finally Theorem C.1 in Tzen and Raginsky [2019b] will hold true in our setting, with the modified choice of

$$N = \left\lceil \left( \frac{C\sqrt{d}}{\varepsilon} \cdot L((R \vee 1) + \sqrt{2d} + \sqrt{6}) \cdot (16\sqrt{6\pi RdT} + 5\sqrt{\ln 4(d+1)}) \right)^2 \right\rceil, \tag{63}$$

For completeteness we will now restate our adaptation of Theorem C.1.

**Corollary 3.13.** *(Theorem C.1. from Tzen and Raginsky [2019b])*

*For any $\varepsilon > 0$ and any $R > 0$, there exist $N = \mathrm{poly}(1/\varepsilon, d, L, R, T)$ points $z_1, \ldots, z_N \in \mathbb{R}^d$, for which the following holds:*

$$\max_{n \leq N} \|z_n\| \leq 8\sqrt{(d+6)\ln N}$$

$$\sup_{x \in \mathrm{B}^d(R)} \sup_{t \in [0,1]} \left| \frac{1}{N} \sum_{n=1}^{N} f\left(e^{-t}x + (1 - e^{-2t})^{1/2}z_n\right) - U_t f(x) \right| \leq \varepsilon$$

$$\sup_{x \in \mathrm{B}^d(R)} \sup_{t \in [0,1]} \left\| \frac{1}{N} \sum_{n=1}^{N} \nabla f\left(e^{-t}x + (1 - e^{-2t})^{1/2}z_n\right) - \nabla U_t f(x) \right\| \leq \varepsilon$$

We now have everything that is required to show the neural network approximation results.

*Remark* D.4. The same computation for our tight-bound from Proposition 3.12 leads to:

$$H(\mathcal{G}, F, \varepsilon) \leq \sqrt{2d \ln \left( \frac{e^{-\epsilon^2/2}\sqrt{3TR}}{\epsilon} \right)_+} \tag{64}$$

Moving forward:

$$J(\mathcal{G}, F) \leq \int_0^{\sqrt{W(1)}} \sqrt{2d \ln \left( \frac{e^{-\epsilon^2/2}\sqrt{3TR}}{\epsilon} \right)_+} \, d\epsilon$$

Where $W(1)$ is the solution to $-x = \ln x$. Unfortunately, we weren't able to find a closed-form solution to this integral.

# E   NEURAL NETWORK APPROXIMATION

**Corollary E.1.** *Under Assumption B.2, the vector field $\nabla \ln U_t f(x)$ is bounded in norm by $\frac{L}{c}$ and is Lipschitz with constant $\frac{L}{c} + \frac{L^2}{c^2}$ where L is the max of the Lip constant of $f$ and $\nabla f$.*

*Proof.* By direct application of Lemma B.1. (Tzen and Raginsky [2019b]) and our Lemma 3.6, which assures that OU semi-group commutes with the gradient operator, we have that the results of this Corollary hold. $\qquad \square$

We now proceed to adapt one of the main theorems in Tzen and Raginsky [2019b]. Whilst the changes are minor to the sketch in Tzen and Raginsky [2019a] some are subtle thus we have incorporated this proof for completeness. We highlight in magenta the subtle changes required to adapt the result.

**Corollary E.2.** *(Tzen and Ragisnky)* *Let* $0 < \varepsilon < 4L/c$ *and* $R > 0$ *be given. Then there exists a neural net* $\widehat{v}$ : $\mathbb{R}^d \times [0,1] \to \mathbb{R}^d$ *of size polynomial in* $1/\varepsilon, d, L, R, c, 1/c$, *such that the activation function of each neuron is an element of the set* $\{\sigma, \sigma', \mathrm{ReLU}\}$, *and the following holds:*

$$\sup_{x \in \mathrm{B}^d(R)} \sup_{t \in [0,1]} \|\widehat{v}(x,t) - \nabla \ln U_t f(x)\| \le \varepsilon$$

*and*

$$\max_{i \in [d]} \sup_{x \in \mathbb{R}^d} \sup_{t \in [0,1]} |\widehat{v}_i(x,t)| \le \frac{2L}{c}.$$

*Proof.* Let $\delta = \frac{c^2 \varepsilon}{16L}$. By Theorem C.1 (which has been proved to hold true in our settings in Appendix C), there exist points $z_1, \ldots, z_N \in \mathbb{R}^d$ with $N = \mathrm{poly}(1/\delta, d, L, R)$, such that $R_{N,d} := \max_{n \le N} \|z_n\| \le 8\sqrt{(d+6)\ln N}$, and the function $\varphi : \mathbb{R}^d \times [0,1] \to \mathbb{R}$ defined by

$$\varphi(x,t) := \frac{1}{N} \sum_{n=1}^{N} f\left(e^{-t}x + (1 - e^{-2t})^{1/2} z_n\right) \tag{65}$$

satisfies

$$\sup_{x \in \mathrm{B}^d(R)} \sup_{t \in [0,1]} |\varphi(x,t) - U_t f(x)| \le \delta \quad \text{and} \quad \sup_{x \in \mathrm{B}^d(R)} \sup_{t \in [0,1]} \|\nabla \varphi(x,t) - \nabla U_t f(x)\| \le \delta$$

By Assumption B.4, there exists a neural net $\widehat{f} : \mathbb{R}^d \to \mathbb{R}$ be that approximates $f$ and the gradient of $f$ to accuracy $\delta$ on the blown-up ball $\mathrm{B}^d(R + R_{N,d})$. Then the function

$$\widehat{\varphi} : \mathbb{R}^d \times [0,1] \to \mathbb{R}, \quad \widehat{\varphi}(x,t) := \frac{1}{N} \sum_{n=1}^{N} \widehat{f}\left(e^{-t}x + (1 - e^{-2t})^{1/2} z_n\right)$$

can be computed by a neural net of size $N \cdot \mathrm{poly}(1/\delta, d, L, R)$, such that

$$\sup_{x \in \mathrm{B}^d(R)} \sup_{t \in [0,1]} |\widehat{\varphi}(x,t) - U_t f(x)|$$

$$\le \sup_{x \in \mathrm{B}^d(R)} \sup_{t \in [0,1]} |\widehat{\varphi}(x,t) - \varphi(x,t)| + \sup_{x \in \mathrm{B}^d(R)} \sup_{t \in [0,1]} |\varphi(x,t) - U_t f(x)|$$

$$\le \sup_{x \in \mathrm{B}^d(R)} \sup_{t \in [0,1]} \left| \frac{1}{N} \sum_{n=1}^{N} \widehat{f}\left(x + (1 - e^{-2t})^{1/2} z_n\right) - \frac{1}{N} \sum_{n=1}^{N} f\left(x + (1 - e^{-2t})^{1/2} z_n\right) \right|$$

$$+ \sup_{x \in \mathrm{B}^d(R)} \sup_{t \in [0,1]} |\varphi(x,t) - U_t f(x)|$$

$$\le \sup_{x \in \mathrm{B}^d(R + R_{N,d})} |\widehat{f}(x) - f(x)| + \sup_{x \in \mathrm{B}^d(R)} \sup_{t \in [0,1]} |\varphi(x,t) - U_t f(x)| \le 2\delta$$

where the third inequality follows since $e^{-t} \in [0,1]$ and the final inequality follows since

$$\max_{n} \sup_{t \in [0,1]} (1 - e^{-2t})^{1/2} \|z_n\| = \max_{n} \|z_n\| = R_{N,d}$$

Similarly

$$\sup_{x \in \mathrm{B}^d(R)} \sup_{t \in [0,1]} \|\nabla \widehat{\varphi}(x,t) - \nabla U_t f(x)\|$$

$$\le \sup_{x \in \mathrm{B}^d(R)} \sup_{t \in [0,1]} \|\nabla \widehat{\varphi}(x,t) - \nabla \varphi(x,t)\| + \sup_{x \in \mathrm{B}^d(R)} \sup_{t \in [0,1]} \|\nabla \varphi(x,t) - \nabla U_t f(x)\|$$

$$\le \sup_{x \in \mathrm{B}^d(R + R_{N,d})} \|\nabla \widehat{f}(x) - \nabla f(x)\| + \sup_{x \in \mathrm{B}^d(R)} \sup_{t \in [0,1]} \|\nabla \varphi(x,t) - \nabla U_t f(x)\| \le 2\delta.$$

Since $f$ is $L$-Lipschitz and bounded below by $c$, we have $U_t f(x) \geq \mathbb{E}_{Z \sim \mathcal{N}(0,I)}[c] = c$, and

$$U_t f(x) = \mathbb{E}_{Z \sim \mathcal{N}(0,I)}\left[f(e^{-t}x + (1 - e^{-2t})^{1/2}Z)\right] \leq \mathbb{E}_{Z \sim \mathcal{N}(0,I)}\left[L(\|x\| + \sqrt{2}\|z\|) + f(0)\right]$$
$$= L\|x\| + f(0) + L\sqrt{2}\mathbb{E}[\|z\|]$$
$$\leq L(\|x\| + \sqrt{2d}) + f(0)$$

Thus it follows that $c \leq U_t f(x) \leq L(\|x\| + \sqrt{2d}) + f(0)$ for any $x \in \mathbb{R}^d$ and $t \in [0,1]$. Therefore, on $\mathrm{B}^d(R) \times [0,1]$,

$$\frac{c}{2} \leq \widehat{\varphi}(x,t) \leq L(R + \sqrt{2d}) + f(0) + \frac{c}{2}$$

where we use $\delta \leq c/4$. Without loss of generality, we may assume that $L \geq 1$. Then, for any $x \in \mathrm{B}^d(R)$ and $t \in [0,1]$

$$\|\nabla \ln \widehat{\varphi}(x,t) - \nabla \ln U_t f(x)\|$$
$$= \left\|\frac{\nabla \widehat{\varphi}(x,t)}{\widehat{\varphi}(x,t)} - \frac{\nabla U_t f(x)}{U_t f(x)}\right\|$$
$$\leq \frac{1}{\widehat{\varphi}(x,t)}\|\nabla \widehat{\varphi}(x,t) - \nabla U_t f(x)\| + \left\|\frac{\nabla U_t f(x)}{U_t f(x)}\right\|\frac{|\widehat{\varphi}(x,t) - U_t f(x)|}{\widehat{\varphi}(x,t)}$$
$$\leq \frac{2L}{c} \cdot 2\delta + \frac{L}{c} \cdot \frac{2}{c} \cdot 2\delta$$
$$\leq \frac{\varepsilon}{2},$$

where we have used Corollary E.1 to bound $\left\|\frac{\nabla U_t f}{U_t f}\right\| \leq L/c$. In other words, $\nabla \ln \widehat{\varphi}(x,t)$ approximates $\nabla \ln U_t f(x)$ to accuracy $\varepsilon/2$ uniformly on $\mathrm{B}^d(R) \times [0,1]$. It remains to approximate $\nabla \ln \widehat{\varphi}(x,t)$ by a neural net to accuracy $\varepsilon/2$.

To that end, we first represent $\nabla \ln \widehat{\varphi}(x,t)$ as a composition of several elementary operations and then approximate each step by a neural net. Specifically, the computation of $v_i = \partial_i \ln \widehat{\varphi}(x,t)$ can be represented as a computation graph with the following structure:

1. Compute $a = \widehat{\varphi}(x,t)$.
2. Compute $b_i = \partial_i \widehat{\varphi}(x,t)$.
3. Compute $r = 1/a$.
4. Compute $v_i = rb_i$.

Given $x$ and $t$, $a$ is computed by a neural net with activation function $\sigma$, of size $\mathrm{poly}(1/\delta, d, L, R)$ and depth poly $(1/\delta, d, L, R)$. Therefore, by the cheap gradient principle (Lemma D.1 from Tzen and Raginsky [2019b]), $b_i$ can be computed by a neural net of size poly $(1/\delta, d, L, R)$, where the activation function of each neuron is an element of the set $\{\sigma, \sigma'\}$. Next, since $a$ takes values in $[c/2, L(R + \sqrt{2d}) + f(0) + c/2]$, by Lemma D.2 from Tzen and Raginsky [2019b] the reciprocal $r = 1/a$ can be computed to accuracy $\varepsilon/(4L\sqrt{d})$ by a 2-layer neural net with activation function $\sigma$ and of size

$$\mathcal{O}\left(\frac{4}{c^2} \cdot (L(R + \sqrt{2d}) + f(0) + c/2) \cdot \frac{4L\sqrt{d}}{\varepsilon}\right) \leq \mathrm{poly}(1/\varepsilon, d, L, R, c, 1/c)$$

Let $\widehat{r}$ denote the resulting approximation. Then, since $|b_i| \leq 2L$ and $|\widehat{r}| \leq 2/c + \varepsilon/(4L\sqrt{d}) \leq 4/c$, by Lemma D.2 the product $\widehat{r}b_i$ can be approximated to accuracy $\varepsilon/4\sqrt{d}$ by a 2-layer neural net with activation function $\sigma$ and with at most

$$\mathcal{O}\left((4/c \vee 2L)^2 \cdot \frac{4\sqrt{d}}{\varepsilon}\right) \leq \mathrm{poly}(1/\varepsilon, d, L, 1/c)$$

neurons. The overall accuracy of the approximation is

$$|\widehat{v}_i - v_i| \leq |\widehat{v}_i - \widehat{r}b_i| + |\widehat{r}b_i - rb_i| \leq \frac{\varepsilon}{2\sqrt{d}}$$

Thus, the vector $v = (v_1, \ldots, v_d)$ can be $\varepsilon/2$-approximated by $\tilde{v}(x,t)$, where $\tilde{v} : \mathbb{R}^d \times [0,1] \to \mathbb{R}^d$ is a neural net with vector-valued output that has the size $\mathrm{poly}(1/\varepsilon, d, L, R, c, 1/c)$. Finally, since $\sup_{x \in \mathrm{B}^d(R)} \sup_{t \in [0,1]} |\tilde{v}_i(x,t)| \leq 2L/c$, the function

$$\widehat{v}_i(x,t) := \min\left\{\max\left\{\tilde{v}_i(x,t), -2L/c\right\}, 2L/c\right\}$$

is continuous, takes values in $[-2L/c, 2L/c]$ and coincides with $\tilde{v}_i$ on $\mathrm{B}^d(R) \times [0,1]$. Moreover, the min and max operations can each be implemented exactly using $\mathcal{O}(1)$ ReLU neurons. $\square$

**Corollary 3.1.** *Suppose Assumptions 1-3 are in force. Let $L$ denote the maximum of the Lipschitz constants of $f$ and $\nabla f$. Then for all $0 < \epsilon < 16L^2/c^2$, there exists a neural net $\hat{v} : R^d \times [0,1] \to R^d$ with size polynomial in $1/\epsilon, d, L, c, 1/c$ such that the activation function of each neuron in the set of $\{\sigma, \sigma', ReLU\}$, and the following hold: If $\{\hat{x}_t\}_{t \in [0,1]}$ is the diffusion process governed by the Itô SDE:*

$$d\hat{x}_t = \hat{b}(\hat{x}_t, t)dt + \sqrt{2\beta}dW_t \tag{66}$$

*with $x_0 \sim p_1 \approx \mathcal{N}(0, I)$ with the drift $\hat{b}(x,t) = -(x - 2\hat{v}(x, 1-t))$, then $\hat{\mu} := \mathrm{Law}(\hat{x}_1)$, satisfies $D(\mu\|\hat{\mu}) \leq \epsilon$.*

*Proof.* For any $R > 0$, Corollary E.2 guarantees the existence of a neural net $\widehat{v} : \mathbb{R}^d \times [0,1] \to \mathbb{R}^d$ that satisfies

$$\sup_{x \in \mathrm{B}^d(R)} \sup_{t \in [0,1]} \|\widehat{v}(x,t) - \nabla \ln U_t f(x)\| \leq \sqrt{\varepsilon} \tag{67}$$

and

$$\max_{i \in [d]} \sup_{x \in \mathbb{R}^d} \sup_{t \in [0,1]} |\widehat{v}_i(x,t)| \leq \frac{2L}{c}. \tag{68}$$

Let $\boldsymbol{\mu} := \mathrm{Law}\left(x_{[0,1]}\right)$ and $\widehat{\boldsymbol{\mu}} := \mathrm{Law}\left(\widehat{x}_{[0,1]}\right)$. The Girsanov formula gives

$$\mathrm{KL}(\boldsymbol{\mu}\|\widehat{\boldsymbol{\mu}}) = \frac{1}{2}\int_0^1 \mathbf{E}\left\|b\left(x_t, t\right) - \widehat{b}\left(x_t, t\right)\right\|^2 \mathrm{d}t$$

where the interchange of the integral and the expectation follows from Fubini's theorem because both $b$ and $\widehat{b}$ are bounded by Corollary E.1 and (68). We now proceed to estimate the integrand. For each $t \in [0,1]$

$$\mathbf{E}\left\|b\left(x_t, t\right) - \widehat{b}\left(x_t, t\right)\right\|^2$$
$$= \mathbf{E}\left[\left\|b\left(x_t, t\right) - \widehat{b}\left(x_t, t\right)\right\|^2 \cdot \mathbf{1}\left\{x_t \in \mathrm{B}^d(R)\right\}\right] + \mathbf{E}\left[\left\|b\left(x_t, t\right) - \widehat{b}\left(x_t, t\right)\right\|^2 \cdot \mathbf{1}\left\{x_t \notin \mathrm{B}^d(R)\right\}\right]$$
$$=: T_1 + T_2,$$

where $T_1 \leq \varepsilon$ by (68). To estimate $T_2$, we first observe that, since the OU drift is bounded in norm by $L/c$ by E.1, we have

$$\mathbf{P}\left\{\sup_{t \in [0,1]} \|x_t\| \geq R\right\} \leq \frac{\sqrt{d} + L/c}{R}$$

(Bubeck et al. [2018], Lemma 3.8). Therefore,

$$T_2 \leq \frac{9dL^2}{c^2} \cdot \frac{\sqrt{d} + L/c}{R}$$

Since some of the bounds differ from the original Tzen and Raginsky [2019b] we verify that the bound still holds for our

drift. We used that $d \geq 2$.

$$T_2 = \mathbf{E}\left[\left\|b\left(x_t, t\right) - \widehat{b}\left(x_t, t\right)\right\|^2 \cdot \mathbf{1}\left\{x_t \notin \mathrm{B}^d(R)\right\}\right] = \int_{x_t \notin \mathrm{B}^d(R)} \|b\left(x_t, t\right) - \widehat{b}\left(x_t, t\right)\|^2 dP_{x_t} =$$

$$= \int_{x_t \notin \mathrm{B}^d(R)} 2\|b\left(x_t, t\right)\|^2 + 2\|\widehat{b}\left(x_t, t\right)\|^2 dP_{x_t} \leq \int_{x_t \notin \mathrm{B}^d(R)} 2\|b\left(x_t, t\right)\|^2 + 2d\left(\frac{2L}{c}\right)^2 dP_{x_t} \leq$$

$$\leq \int_{x_t \notin \mathrm{B}^d(R)} 2\|\nabla \ln U_t f(x_t)\|^2 + 8d\left(\frac{L}{c}\right)^2 dP_{x_t} = \int_{x_t \notin \mathrm{B}^d(R)} 2\left\|\frac{\nabla U_t f(x_t)}{U_t f(x_t)}\right\|^2 + 8d\left(\frac{L}{c}\right)^2 dP_{x_t} \leq$$

$$\leq \int_{X_t \notin \mathrm{B}^d(R)} 2\frac{L^2}{c} + 8d\left(\frac{L}{c}\right)^2 dP_{x_t} \leq 9d\frac{L^2}{c^2} P\left\{\sup_{t \in [0,1]} \|x_t\| \geq R\right\} \leq \frac{9dL^2}{c^2} \cdot \frac{\sqrt{d} + L/c}{R}$$

Choosing $R$ large enough to guarantee $T_2 \leq \varepsilon$ and putting everything together, we obtain $D(\boldsymbol{\mu}\|\widehat{\boldsymbol{\mu}}) \leq \varepsilon$. Therefore, $D(\mu\|\widehat{\mu}) \leq D(\boldsymbol{\mu}\|\widehat{\boldsymbol{\mu}}) \leq \varepsilon$ by the data processing inequality. □

Finally, we would like to highlight what happens when we sample $\hat{x}_0 \sim \mathcal{N}(0, 1)$ rather than $p_T$. Whilst our results are done for $t \in [0, 1]$ one can see that the overall approximation results will hold for $t \in [0, T]$.

**Remark 3.2.** *Assuming $\pi$ satisfies a logarithmic Sobolev inequality we extend the time domain to $t \in [0, T]$ and sampling $\hat{x}_0 \sim \mathcal{N}(0, I)$ approximately, it follows that $D(\mu\|\hat{\mu}) \leq e^{-T}\mathrm{KL}(\pi\|\mathcal{N}(0, 1)) + T\epsilon$*

*Proof.* First, we remark that the estimation results and the results in Corollary 3.1 apply to the $t \in [0, T]$ setting, however, they will introduce a polynomial dependency in $T$ for the size of the network.

As in the above proof, we apply the Girsanov theorem to control the path KL, however here, the starting distributions of the two Ito processes are no longer the same thus, we get an extra term from the chain rule:

$$\mathrm{KL}(\boldsymbol{\mu}\|\widehat{\boldsymbol{\mu}}) = \mathrm{KL}(p_T\|\mathcal{N}(0, 1)) + \frac{1}{2}\int_0^T \mathbf{E}\left\|b\left(x_t, t\right) - \widehat{b}\left(x_t, t\right)\right\|^2 \mathrm{d}t \tag{69}$$

$$\leq \mathrm{KL}(p_T\|\mathcal{N}(0, 1)) + T\epsilon \tag{70}$$

$$\leq e^{-T}\mathrm{KL}(\pi\|\mathcal{N}(0, 1)) + T\epsilon \tag{71}$$

Where the final inequality follows from Theorem 5.2.1 in Bakry et al. [2014] under the assumption that $\pi$ satisfies a log-Sobolev inequality. This completes the circle and fully extends Theorem 3.1 from Tzen and Raginsky [2019b] to our denoising diffusion setting.

□

Finally, note that if we assume that $\mathrm{supp}\,\pi \subseteq \mathrm{B}^d(R)$ from Theorem 2 of Chen et al. [2022] it follows that:

$$\mathrm{TV}\left(\mathrm{Law}\hat{x}_t, \pi\right) \leq \mathcal{O}\left(\sqrt{\mathrm{KL}\left(\pi\|\mathcal{N}(0, I)\right)}\exp(-T) + \epsilon\sqrt{T}\right). \tag{72}$$

This result complements Corollary 3.1 very nicely as unlike Chen et al. [2022] we no longer require assuming an $\epsilon$ error on the score but instead prove such error can be attained.

# F PBM TRANSITION DENSITY

As PBM is a linear SDE we know its transition densities are Gaussian thus finding its first and second moments fully determines it.

## F.1 MEAN

Taking expectations on the solution to the PBM-SDE yields an ODE for the mean of the transition density:

$$\frac{d\mu_t}{dt} = \left(\frac{d\alpha_t}{dt}\right)\frac{\mu_t}{\alpha_T - \alpha_t}$$

separating variables:

$$\frac{1}{\mu_t}d\mu_t = \frac{1}{\alpha_T - \alpha_t}d\alpha_t$$

integrating both sides:

$$\ln\frac{\mu_t}{\mu_s} = \ln\frac{\alpha_T - \alpha_t}{\alpha_T - \alpha_s}$$

thus:

$$\mu_t = \mu_s\frac{\alpha_T - \alpha_t}{\alpha_T - \alpha_s}$$

and at $s = 0$:

$$\mu_t = x\frac{\alpha_T - \alpha_t}{\alpha_T - \alpha_0}$$

## F.2   VARIANCE

Applying Ito's Lemma to the PBM-SDE $z_t = x_t^2$ yields,

$$dz_t = \left(-\left(\frac{d\alpha_t}{dt}\right)\frac{2z_t}{\alpha_T - \alpha_t} + \left(\frac{d\alpha_t}{dt}\right)\right)dt + 2\left(\frac{d\alpha_t}{dt}\right)^{1/2}x_t dW_t \tag{73}$$

taking expectations and using the martingale property we have:

$$\frac{d\mu_z(t)}{dt} = \left(\frac{d\alpha_t}{dt}\right)\left(1 - \frac{2\mu_z(t)}{\alpha_T - \alpha_t}\right) \tag{74}$$

As before let us compute the integrating factor :

$$e^{-\int_s^t\left(\frac{d\alpha_\tau}{d\tau}\right)\frac{2Z_\tau}{\alpha_T - \alpha_\tau}d\tau} = \left(\frac{\alpha_T - \alpha_s}{\alpha_T - \alpha_t}\right)^2$$

thus:

$$\left(\frac{d\alpha_t}{dt}\right)\left(\frac{\alpha_T - \alpha_s}{\alpha_T - \alpha_t}\right)^2 = \frac{d(((\alpha_T - \alpha_s)^2/(\alpha_T - \alpha_t)^2))\mu_z(t))}{dt} \tag{75}$$

$$\int_s^t\left(\frac{d\alpha_\tau}{d\tau}\right)\left(\frac{\alpha_T - \alpha_s}{\alpha_T - \alpha_\tau}\right)^2 d\tau = \left(\frac{\alpha_T - \alpha_s}{\alpha_T - \alpha_t}\right)^2\mu_z(t) + \mu_z(s) \tag{76}$$

$$\int_s^t\left(\frac{\alpha_T - \alpha_s}{\alpha_T - \alpha_\tau}\right)^2 d\alpha_\tau = \left(\frac{\alpha_T - \alpha_s}{\alpha_T - \alpha_t}\right)^2\mu_z(t) + \mu_z(s) \tag{77}$$

$$(\alpha_T - \alpha_s)^2\left(\left(\frac{1}{\alpha_T - \alpha_t}\right) - \left(\frac{1}{\alpha_T - \alpha_s}\right)\right) = \left(\frac{\alpha_T - \alpha_s}{\alpha_T - \alpha_t}\right)^2\mu_z(t) + \mu_z(s) \tag{78}$$

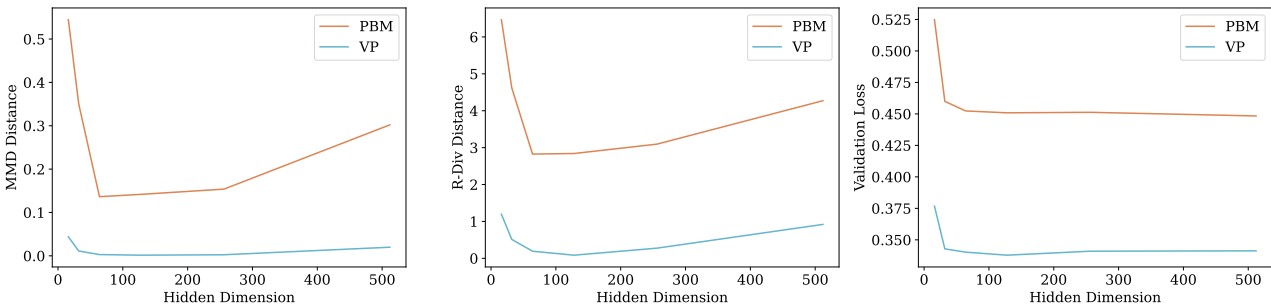

(a) Distances between $\pi$ and $p_\theta^{\mathrm{model}}$ at time $T$ over a 10-dimensional Funnel, with results obtained from 3 different seeds. The $x$-axis represents various hidden layer dimensions

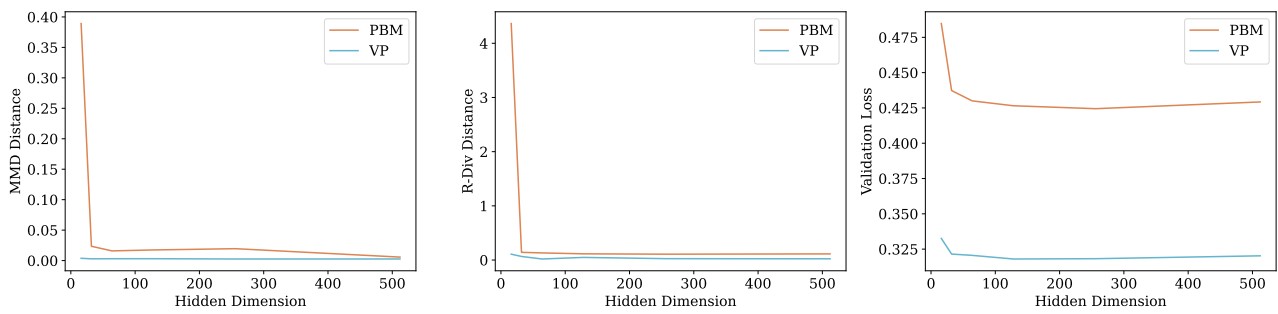

(b) Distances between $\pi$ and $p_\theta^{\mathrm{model}}$ at time $T$ over a 10-dimensional GMM-10, with results obtained from 3 different seeds. The $x$-axis represents various hidden layer dimensions

rearranging:

$$(\alpha_T - \alpha_t) - \frac{(\alpha_T - \alpha_t)^2}{\alpha_T - \alpha_s} - \left(\frac{\alpha_T - \alpha_t}{\alpha_T - \alpha_s}\right)^2 \mu_z(s) = \mu_z(t) \tag{79}$$

Now using $\mathrm{var}(X) = \mathbb{E}[X^2] - \mathbb{E}[X]^2$ give the desired result.

$$\mathrm{var}(z_t) = (\alpha_T - \alpha_t) - \frac{(\alpha_T - \alpha_t)^2}{\alpha_T - \alpha_s} = \frac{(\alpha_T - \alpha_t)(\alpha_t - \alpha_s)}{\alpha_T - \alpha_s} \tag{80}$$

### F.3 TRANSITION DENSITY

Using the results from the previous two sections and $s = 0$ we have:

$$p(x_t|x_0) = \mathcal{N}\left(x_t \middle| \frac{\alpha_T - \alpha_t}{\alpha_T - \alpha_0} x_0, \frac{(\alpha_T - \alpha_t)(\alpha_t - \alpha_0)}{\alpha_T - \alpha_0}\right).$$

## G SINTHETIC EXPERIMENTAL DETAILS

We employed a neural network architecture consisting of 5 MLP layers ReLU activation functions and dropout set to 0. The learning rate was set to 0.00001, and we conducted training over 100 epochs for each model in our study. We utilized the Adam optimizer along with a LambdaLR scheduler.

The datasets were divided into training, validation, and testing sets. The training set consisted of 100,000 samples, the validation set consisted of 20,000 samples, and the testing set consisted of 10,000 samples. To evaluate model performance, we computed Maximum Mean Discrepancy (MMD) and r-divergence between samples generated by the trained model at the final time step and samples from the testing set.

### G.1 NOISE SCHEDULE

For both VP-SDE and PBM-SDE we use the following linear noise schedule:

$$\beta_t = \frac{\mathrm{d}\alpha_t}{\mathrm{d}t} = \beta_{\min}\frac{(T-t)}{T} + \beta_{\max}\frac{t}{T} \tag{81}$$

with $T = 1$, $\beta_{\min} = 0.1$, $\beta_{\max} = 20$.

### G.2 SCORE ESTIMATION ACROSS NETWORK SIZE

For this experiment, we fixed $d = 10$, and varied the network width across $4, 16, 32, 64, 128, 256, 512$ for both GMM-10 and Funnel. The results obtained are in line with those in Section 4.1.2.

### G.3 MMD AND R-DIVERGENCE DETAILS

We use MMD-Fuse [Biggs et al., 2023][3] codebase to compute MMD and a Laplace kernel.

For the R-divergence, we use a standard Gaussian kernel and a Scott bandwidth estimator [Scott, 1979] using the Scipy library [Virtanen et al., 2020].

## H  IMAGE DATASETS EXPERIMENTAL DETAILS

The used network was a convolutional neural network (CNN) tailored for image processing tasks, specifically optimized for images sized $28 \times 28$ pixels with three color channels (RGB). It consisted of multiple layers of convolutional and residual blocks, featuring a total of 32 channels within the convolutional layers and incorporating one residual block. The architecture integrated the principle of channel multiplication, sequentially scaling the number of channels in each layer by factors of $1, 2$, and $2$. The model is also utilizing residual blocks for both upscaling and downscaling operations. The total number of parameters is $1.34$ million.

### H.1 NOISE SCHEDULE

For both VP-SDE and PBM-SDE we use the following linear noise schedule:

$$\beta_t = \frac{\mathrm{d}\alpha_t}{\mathrm{d}t} = \beta_{\min}\frac{(T-t)}{T} + \beta_{\max}\frac{t}{T}, \tag{82}$$

with values:

- for VP-SDE: $T = 1$, $\beta_{\min} = 0.0001$, $\beta_{\max} = 20$.
- for PBM: $T = 2$, $\beta_{\min} = 0.001$, $\beta_{\max} = 1$.

### H.2 FID DETAILS

We use Jiralerspong et al. [2023][4] codebase to compute the FID metric over the test dataset, employing a sample size of 1000 and the CLIP feature extractor.

### H.3 SAMPLES

---

[3]https://github.com/antoninschrab/mmdfuse
[4]https://github.com/marcojira/fld

VP-SDE                                    PBM

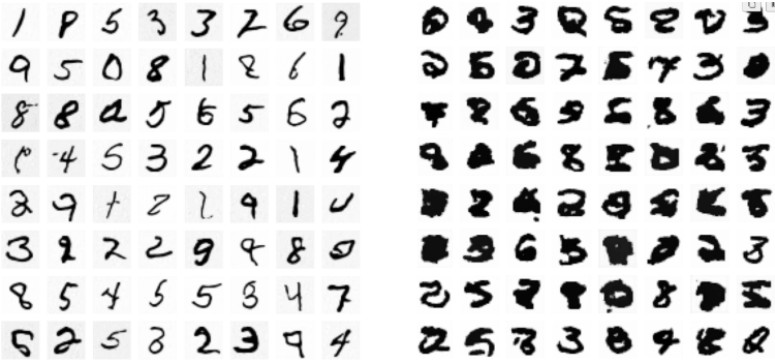

Figure 6: MNIST samples

VP-SDE                                    PBM

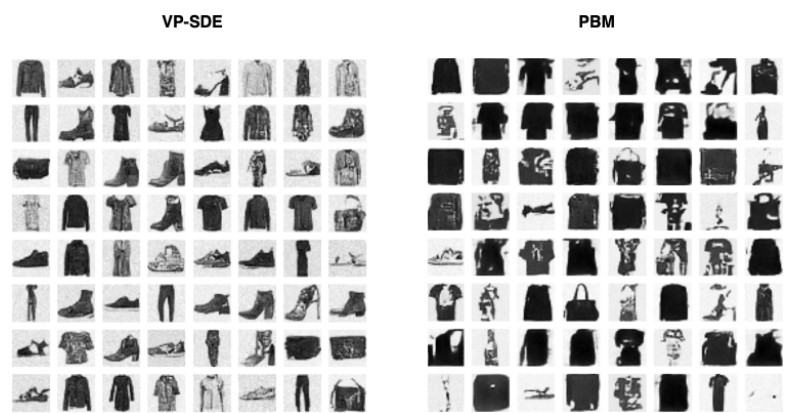

Figure 7: Fashion-MNIST samples

VP-SDE                                    PBM

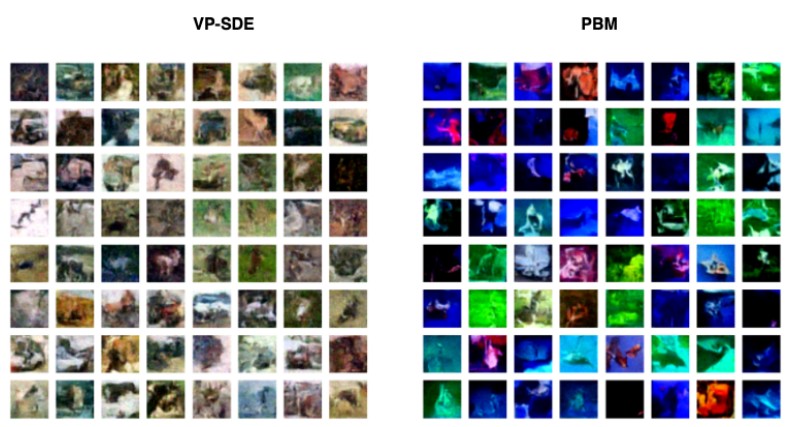

Figure 8: CIFAR-10 samples