# OpenReview forum: "To smooth a cloud or to pin it down: Expressiveness guarantees and insights on score matching in denoising diffusion models"
_auai.org/UAI/2024/Conference — UAI 2024 poster_

### Official Review · Reviewer_3pkw · 2024-03-06

**Q2-1 Originality-Novelty:** 3
**Q2-2 Correctness-Technical Quality:** 3
**Q2-5 Clarity Of Writing:** 2

**Q1 Summary And Contributions:**

This paper offers a theoretical analysis to quantify the estimation error of the score function used in DDPM and VP-SDE based on the neural networks. The author introduces various stochastic control theories to prove VP-SDE is better than pinned Brownian motion-based methods.

**Q2-3 Extent To Which Claims Are Supported By Evidence:**

3: Good: the main claims are supported by convincing evidence (in the form of adequate experimental evaluation, proofs, (pseudo-)code, references, assumptions).

**Q2-4 Reproducibility:**

3: Good: key resources (e.g. proofs, code, data) are available and key details (e.g. proofs, experimental setup) are sufficiently well-described for competent researchers to confidently reproduce the main results.

**Q3 Main Strengths:**

1. This paper shows that the neural network can precisely estimate the score function based on VP-SDE, which has potential value.
2. The experimental results enhance the theory for this paper.

**Q4 Main Weakness:**

1. This paper is hard to follow due to the writing mistakes. For example, 1) the OU (Ornstein–Uhlenbeck) process first shows in the Abstract without the full name until Sec. 3. 2) Most contents in the Introduction are directly from Vargas et al. [1] without necessary citations. 3) Mathematical symbols are confused. For instance, the symbol in Eq. 3. $y_{0} \sim p_{T}$ and thus $p_{T}$ has been a known distribution. However, following the contents show that $y_{0} \sim p_{T}(y_{0})$. This shows that we need first to get $y_{0}$ and use it as the variable to calculate $p_{T}$, which creates a conflict. 4) Please check the reference since the same paper has been cited twice such as "Diffusion Schrödinger bridge with applications to score-based generative modeling".

2. The setting for this paper is confusing. The author claims that they use the Denoising diffusion samplers (DDS) theory as the basis to analyze the estimation error for VP-SDE. However, DDS designs for the unnormalized probability density functions, which is different from the VP-SDE. If this paper has the same setting as VP-SDE, the sample paths could be calculated directly. In this condition, the probability space of VP-SDE satisfies the definition proposed by Tzen et al. and we could directly calculate the estimation error based on the target diffusion process.

3. The main contribution of this paper is confusing since Vargas et al. and Tzen et al. [2] have finished almost all the theory and proof. Then, the author does not show why using the theory of Tzen et al. to analyze the VP-SDE is difficult. At least, the author should clarify what assumption has been broken by VP-SDE under the framework of Tzen et al..


[1] Vargas et al., Denoising diffusion samplers. ICLR 2023.

[2] Tzen et al., Theoretical guarantees for sampling and inference in generative models with latent diffusions. COLT 2019.

**Q5 Detailed Comments To The Authors:**

The author should clarify the setting problems. Following a concrete setting, what assumption in Tzen et al. has been broken? In the end, the author should emphasize how DDS helps the VP-SDE be possible to back to the framework of Tzen et al..

**Q9 Complying With Reviewing Instructions:**

Yes

---

> ### Author Rebuttal · Authors · 2024-04-02
>
> Thank you for the helpful and insightful comments. Following the guidance email from UAI, we are allowed to link anonymised links; thus, to facilitate the discussion we have uploaded a **revised manuscript** to an anonymised repository:
>
> Link:   https://anonymous.4open.science/r/rebuttal-A7C2/UAI.pdf
>
> We have coloured changes addressing only your requests in **orange**.We have also detailed some of the added paragraphs and sentences in this rebuttal:
>
> >  … which creates a conflict.
>
> We have removed instances of the notation $y_0 \sim p_T(y_0)$. As you pointed out, this is semantically wrong. In cases where we want to refer to the form of $p_T$ explicitly, we now use the shorthand $p_T \approx N(0, \sigma^2 I)$ or use a dummy variable.
>
> > The setting for this paper is confusing …
>
> In addition to providing an sampling algorithm, DDS [1] establishes a connection between the time reversal of VP-SDEs and stochastic control. From the connections in [1], we can establish Remark 3.5, pointing us to Tzen and Raginsky 2019.
>
> Remark 3.5 allowed us to realise that the proof strategy presented in Tzen and Raginsky 2019 could potentially be extended to denoising diffusion models. Their setup is based on the stoch control formulation of sampling rather than time reversals. Thanks to Remark 3.5 and the insights from the DDS paper, we can view stoch control and time reversal as equivalent.
>
> For the reasons above, we have introduced the stoch control setup presented in DDS, as it allows us to pursue the general sketch strategy in Tzen and Raginsky 2019. To clarify this, we have added the following paragraph:
>
> ```To relate denoising diffusion models to the methodology in Tzen and Raginsky 2019, we will introduce connections presented in [1], which relate score matching in VP-SDEs to stochastic control, thus enabling our main results.```
>
> We were then able to start following and adapting the strategy from Tzen and Raginsky 2019 to denoising diffusion models; however, we ran into an issue right away when we noticed that the terminal cost $\bar{g}_{t,x}(z)$ does not satisfy a Lipstchitz condition with a Euclidean metric thus the results on the Pollard entropy and covering numbers we require no longer apply.
>
>
> > The main contribution of this paper is confusing ...
>
> We will break this into a series of arguments:
> 1. Firstly, [1]  simply prompts us towards Remark 3.5, which allows us to view time reversal in denoising diffusions under the lens of stochastic control. Unfortunately, this did not allow us to apply results from  Tzen and Raginsky 2019 directly, as the regularity properties for the OU semi-group and the Heat semi-group are significantly different, thus, their entropy integral bounds no longer apply.
>
> 2. The quantity  $\bar{g}_{t,x}$ for the heat semi-group; is Lipstchitz relative to the Euclidean metric. This is not the case for OU, instead, we can obtain a Lipstchitz property with respect to a different metric:
>
> $$\rho_{O U}\left((t, x),\left(t^{\prime}, x^{\prime}\right)\right)=|| e^{-t} x-x^{\prime} e^{-t^{\prime}}\| \|+\left|t-t^{\prime}\right|^{1 / 2}$$
>
> 3. Once we have a Lipstchitz property, the next step is obtaining a covering number for the space of functions induced by $\bar{g}_{t,x}(z)$  (indexed by $x,t$. Unfortunately, as the Lipchitz property is not Euclidean, we can no longer use the simple cover Lemma used in Tzen and Raginsky 2019 :
>
> $$N\left(\mathcal{G}, L^2(Q), \varepsilon\|F\|_{L^2(Q)}\right) \leq N\left(\mathrm{~B}^d(R),\|\cdot\|, \varepsilon / 2\right) \cdot N\left([0,1],|\cdot|, \varepsilon^2 / 4\right)$$
>
> This standard result known for covers of function spaces with a Euclidean Lipstchitz property [2], does not apply to our metric space.
>
> 4. We then bind the cover for $([0, T] \times B^d(R), \rho_{O U})$ ourselves, which required several non-trivial steps (see the proof for Lemma 3.10). A large bulk of our work was spent on this challenging step.
>
> $$N\left([0, T] \times B^d(R), \rho_{O U}, \epsilon\right) \leq  N\left([0, T],|\cdot|, \epsilon^2 / 4\right) N\left(B^d(R),\|\cdot\|, \epsilon / 2\right)$$
>
> 5. Once we obtained (4.), the bounds derived were slightly different, so a bit of symbol pushing and further simpler derivation steps were required to achieve the final results.
> In short, step 4, was a challenging and differentiating step from the work in Tzen and Raginsky 2019. Furthermore, to the best of our knowledge, our work is the first to derivate these mathematical properties of the OU semi-group.
> We have added the following paragraph to the revised manuscript to detail this:
>
> ```Contrary to tzen2019neural  $g(e^{-t}x + (1-e^{-2t})^{1/2}z)$ where  $x\in B^d(R), z\in \mathbb{R}^d$ is not Lipschitz in a Euclidean sense. As a result, the standard Euclidean covering-number properties used in tzen2019neural no longer apply, and thus we have to derive bounds for these quantities from scratch.```
>
> [1] Denoising diffusion samplers
>
> [2] Understanding machine learning: From theory to algorithms

---

### Official Review · Reviewer_1Jkd · 2024-03-25

**Q2-1 Originality-Novelty:** 2
**Q2-2 Correctness-Technical Quality:** 3
**Q2-5 Clarity Of Writing:** 2

**Q1 Summary And Contributions:**

The paper studies the universal approximation of neural networks to score functions in diffusion models.

**Q2-3 Extent To Which Claims Are Supported By Evidence:**

3: Good: the main claims are supported by convincing evidence (in the form of adequate experimental evaluation, proofs, (pseudo-)code, references, assumptions).

**Q2-4 Reproducibility:**

3: Good: key resources (e.g. proofs, code, data) are available and key details (e.g. proofs, experimental setup) are sufficiently well-described for competent researchers to confidently reproduce the main results.

**Q3 Main Strengths:**

The paper discuss the connection between diffusion models and stochastic control. The convergence guarantees and score estimation results are established.

**Q4 Main Weakness:**

1. The writing is poor. I do not understand the main problem the authors want to study. I only find a lot of results. The logic connection is loose.
2. The contributions are not clear. Many results are from Tzen and Raginsky [2019b]. The authors should specify the position of this work in the literature.
3. The literature review is not comprehensive. The covering number is also studied in https://arxiv.org/abs/2302.07194 and https://arxiv.org/abs/2303.01861. In the non-parametric settings, very recent works https://arxiv.org/abs/2402.07747 and https://arxiv.org/abs/2402.15602 studied the minimax optimality of score estimation. The GD analysis can be found in https://arxiv.org/abs/2401.15604. The authors should discuss how their work is different from the existing literature.

**Q5 Detailed Comments To The Authors:**

NA

**Q9 Complying With Reviewing Instructions:**

Yes

---

> ### Author Rebuttal · Authors · 2024-04-04
>
> We thank the reviewer for their comment. We have now changed the manuscript and we hope our contribution is more clear. We will now address your concern.
>
> >  Many results are from Tzen and Raginsky [2019b].
>
> Our primary findings diverge from those of Tzen and Raginsky [2019b] due to differences in the problem's setup. Initially, [1] directs us to Remark 3.5, which permits us to analyze time reversal in denoising diffusions through the lens of stochastic control. However, this alone does not enable us to directly apply the results from Tzen and Raginsky 2019. This is primarily because the regularity characteristics of the Ornstein-Uhlenbeck (OU) semigroup and the Heat semigroup differ significantly, rendering their entropy integral bounds inapplicable.
>
> The function $\bar{g}_{t,x}(z)$ for the Heat semigroup is Lipschitz with respect to the Euclidean metric. Conversely, for the OU semigroup, we can establish a Lipschitz property with respect to a distinct metric:
>
> $$\rho_{OU}\left((t, x),\left(t^{\prime}, x^{\prime}\right)\right)=|| e^{-t} x-x^{\prime} e^{-t^{\prime}} \|+\left|t-t^{\prime}\right|^{1 / 2}$$
>
> Upon deriving a Lipschitz property, the subsequent challenge involved obtaining a covering number for the function space induced by $\bar{g}_{t,x}(z)$ (indexed by $x$ and $t$). Unfortunately, as the Lipschitz property does not adhere to the Euclidean metric, we cannot employ the simple cover Lemma utilized in Tzen and Raginsky 2019:
>
> $$N\left(\mathcal{G}, L^2(Q), \varepsilon\|F\|_{L^2(Q)}\right) \leq N\left([0, T] \times B^d(R),  || . || +|.|^{1/2}, \epsilon\right) \leq N\left(\mathrm{~B}^d(R),\|\cdot\|, \varepsilon / 2\right) \cdot N\left([0,1],|\cdot|, \varepsilon^2 / 4\right)$$
>
> This well-known result applicable to function space covers with an Euclidean Lipschitz property [2] does not extend to our metric space. Consequently, we had to construct the cover for $([0, T] \times B^d(R), \rho_{OU})$ ourselves, necessitating several non-trivial steps (as detailed in the proof for Lemma 3.10):
>
> $$N\left([0, T] \times B^d(R), \rho_{OU}, \epsilon\right) \leq  N\left([0, T],|\cdot|, \epsilon^2 / 4\right) N\left(B^d(R),\|\cdot\|, \epsilon / 2\right)$$
>
> Once we derived this, the obtained bounds differed slightly, necessitating additional manipulation of symbols and further simplification steps to arrive at the final results.
>
> In summary, step 4 constituted a challenging and distinctive aspect of our work compared to Tzen and Raginsky 2019. Furthermore, to our knowledge, our work represents the first derivation of these mathematical properties of the OU semigroup. We have included the following paragraph in the revised manuscript to elaborate on this point.
>
> “Contrary to tzen2019neural  $g_{x,t}(z) = g(e^{-t}x + (1-e^{-2t})^{1/2}z)$ where  $x\in B^d(R), z\in \mathbb{R}^d$ is not Lipschitz in a Euclidean sense. As a result, the standard Euclidean covering-number properties used in tzen2019neural no longer apply, and thus we have to derive bounds for these quantities from scratch.“
>
> **Revised manuscript link**: https://anonymous.4open.science/r/rebuttal-A7C2/UAI.pdf (changes highlighted in **blue** and **orange**)
>
> [1] Denoising diffusion samplers
>
> [2] Understanding machine learning: From theory to algorithms

---

### Official Review · Reviewer_iHDm · 2024-03-27

**Q2-1 Originality-Novelty:** 4
**Q2-2 Correctness-Technical Quality:** 3
**Q2-5 Clarity Of Writing:** 4

**Q1 Summary And Contributions:**

This paper provides valuable theoretical analysis and insights on denoising diffusion probabilistic models (DDPMs), a powerful class of generative models that has achieved state-of-the-art results across many domains. The main contributions are:

- Proving novel regularity properties for the OU semigroup that enable extending previous results on approximating the Föllmer drift with neural networks to the score function in VP-SDE based DDPMs.
- Providing quantitative error bounds for the neural network approximation of the score and the induced sampling error when using the approximated score for sampling.
- Empirical evidence suggesting that the score function for VP-SDEs can be approximated better by neural networks compared to pinned Brownian motion SDEs, corroborating the theoretical insights.
- The theoretical analysis leveraging stochastic control concepts is technically sound and sheds useful light on the expressiveness of DDPMs based on different underlying SDEs. The error bounds give meaningful guarantees for the score approximation and sampling performance. The empirical study validates some of the insights from theory.

Overall, this paper makes a valuable contribution to the theoretical understanding of DDPMs and provides useful guidance for model design choices. The results are derived rigorously and clearly presented. I recommend this paper be accepted. Some minor comments to improve the presentation are included below.

**Q2-3 Extent To Which Claims Are Supported By Evidence:**

4: Excellent: all claims are supported by very convincing evidence (in the form of comprehensive experimental evaluation, rigorous mathematical proofs, detailed (pseudo-)code, precise references, well-motivated and realistic assumptions) and the authors deliver what they promise.

**Q2-4 Reproducibility:**

3: Good: key resources (e.g. proofs, code, data) are available and key details (e.g. proofs, experimental setup) are sufficiently well-described for competent researchers to confidently reproduce the main results.

**Q3 Main Strengths:**

1. Provides rigorous theoretical analysis of denoising diffusion probabilistic models (DDPMs) using concepts from stochastic control theory, establishing novel connections.
2. Proves new regularity properties for the Ornstein-Uhlenbeck semigroup that enable bounding the error of approximating the score function with neural networks.
3. Derives quantitative error bounds for the neural network approximation of the score function and the induced sampling error.
4. Empirical results validate some of the key theoretical insights, suggesting VP-SDEs allow better neural network approximation compared to pinned Brownian motion SDEs.

**Q4 Main Weakness:**

1. Theoretical analysis relies on some assumptions (e.g. log-Sobolev inequality) that may not hold for all data distributions of interest.
2. Empirical study is limited to simulated low-dimensional datasets - more experiments on complex real-world data would further support the findings.
3. Lack of practicality to the diffusion / score-based model community, since the VP-SDE is already being used quite widely.

**Q5 Detailed Comments To The Authors:**

The performance in Funnel does not seem to improve much with more hidden dimensions for PBM. Why is that?

**Q9 Complying With Reviewing Instructions:**

Yes

---

> ### Author Rebuttal · Authors · 2024-04-07
>
> Dear Reviewer iHDm,
>
> We thank you for your thorough and insightful review and the kind words about our work.  We would like to apologise for the significantly delayed response. We have been working round the clock to improve our manuscript, and we wanted to run further ablations and experiments for you; however, due to the computational demands of training diffusion models, we might not be able to have this ready before the end of the discussion period.
>
> We have now significantly improved the quality of our manuscript (thanks to all the reviews), and the updated version can be found in this anonymised link:
>
> **revised manuscript:** https://anonymous.4open.science/r/rebuttal-A7C2/UAI.pdf
>
> In what follows, we will proceed to discuss and address your questions.
>
> > The performance in Funnel does not seem to improve much with more hidden dimensions for PBM. Why is that?
>
> Despite our efforts in tuning it, PBM in high dimensions for the Funnel performs poorly. As a result, we do see that the performance increase is somewhat diminishing for large hidden dimensions.
>
> In short, PBM is underperforming in this setting to a level where we don't see massive improvement from increasing the size of the network, as it seems to have plateaued for this target; we believe part of this is due to PBM starting from a delta and thus having a more difficult trade-off between coverage of the data distribution but also concentrating well (i.e. not overestimating the concentration of the Funnel).
>
> > Theoretical analysis relies on some assumptions (e.g. log-Sobolev inequality) that may not hold for all data distributions of interest.
>
> Our main result (the estimation error on the score) does not rely on log Sobolev assumptions; however, in the following remark, we made this assumption as it eases the presentation of this remark.
>
> That said, it is possible to lift this assumption by combining results in [Theorem2, 1] with our main result (which we mention towards the end of Appendix E in the revised manuscript) to arrive at a bound of the form :
>
> $$
> \operatorname{TV}\left(\operatorname{Law} \hat{x}_t, \pi\right) \leq \mathcal{O}(\sqrt{\mathrm{KL}(\pi \| \mathcal{N}(0, I))} \exp (-T)+\epsilon \sqrt{T})
> $$
>
> without requiring the log-Sobolev assumption, the rest of our assumptions are not so restrictive and are typically quite common for this kind of result. However, as the reviewer points out, there are indeed still assumptions, and whilst general, they can be violated.
>
> > Lack of practicality to the diffusion / score-based model community,
>
> Whilst the result is not practical in itself, we believe the fact that VP-SDE is used so much strengthens/motivates our work further in that we try to push the theoretical understanding of a very celebrated and used method further. We hope that some of these theoretical insights can inform method design; in particular, the nice factors arising in the cover of the OU-semigroup do suggest that this is a very nice process in terms of the effort (number of parameters) required to learn it.
>
> > Empirical study is limited to simulated low-dimensional datasets - more experiments on complex real-world data would further support the findings.
>
> For the final version of the manuscript, we aim to include further examples such as MNIST and FashionMNIST; we will aim to upload some of these results ideally tomorrow (before the end of the discussion period).
>
> [1] Chen, S., Chewi, S., Li, J., Li, Y., Salim, A. and Zhang, A.R., 2022. Sampling is as easy as learning the score: theory for diffusion models with minimal data assumptions. arXiv preprint arXiv:2209.11215.

---

### Official Review · Reviewer_Anyj · 2024-03-29

**Q2-1 Originality-Novelty:** 2
**Q2-2 Correctness-Technical Quality:** 3
**Q2-5 Clarity Of Writing:** 2

**Q1 Summary And Contributions:**

The paper presents theoretical expressiveness results for denoising diffusion models/samplers (based on the class of Variance-Preserving SDEs), in a setting where the drift of the reverse process is approximated by a neural network and the target distribution admits a density. The results quantify (1) the error on the neural network approximation of the optimal drift, and (2) the resulting error on the approximated target distribution.

The paper draws its derivations from previous work [1], where similar results were obtained for a different family of generative models / samplers, based on a class of SDEs mapping point masses to target distributions (Föllmer process/Pinned Brownian Motion).
To adapt the proof from [1], the paper provides a link between the VP-SDE score and the OU semigroup   (akin to the Föllmer drift and heat semigroup in [1]) and proves the required regularity properties for the latter.

Finally, numerical simulations are presented.
They compare two classes of generative diffusion models: one based on PBM-SDEs and the other based on VP-SDEs. The models are trained using the score-matching loss. For two distinct target distributions admitting a density (GMM and Funnel), models are compared in terms of how well they approximate (1) the score (through test score-matching loss), and (2) the target distribution (through MMD and r-divergence). The experiment is repeated for increasing model sizes and data dimensions, and models based on VP-SDEs are shown to scale better with these two quantities.


[1] Belinda Tzen and Maxim Raginsky. Theoretical guarantees for sampling and inference in generative models with latent diffusions. In Conference on Learning Theory, 2019

**Q2-3 Extent To Which Claims Are Supported By Evidence:**

2: Fair: the main claims are somewhat supported by evidence (but the experimental evaluation may be weak, or does not match entirely with the claims, important baselines may be missing, proofs contain important ideas but lack rigor, algorithmic details are only discussed superficially, references are imprecise, assumptions are not sufficiently motivated or explicated, etc.).

**Q2-4 Reproducibility:**

3: Good: key resources (e.g. proofs, code, data) are available and key details (e.g. proofs, experimental setup) are sufficiently well-described for competent researchers to confidently reproduce the main results.

**Q3 Main Strengths:**

* Popular class of generative models / samplers: the paper targets the very popular class of denoising diffusion models, for which theoretical insights can potentially be of interest to many in the community;
* Connection between diffusion models and stochastic optimal control: while this paper is not the first to do this connection, connecting ideas from two distinct fields (diffusion models and stochastic optimal control) is interesting;
* Reproducibility:  I could not check the derivations but the paper reads rigorous, and detailed proofs are provided in Appendix. Code for running the numerical simulations is also provided.

**Q4 Main Weakness:**

* Originality-Novelty: without being an expert in the field, it seems to me that the novelty is limited as the paper consists in porting existing results from [1]  to another class of SDEs, following a similar sketch for the different proofs (that being said, this does not mean that the paper does not constitute a valuable contribution);
* Presentation/Readability + Experiments:
  * The presentation of the paper can be improved. I think that it could helpful to state in plain English what the theoretical results / derivations are / do, and what they mean. This would make it easier to follow along, even without grasping all technicalities / notations. For instance, Section 3.4 starts right away with a (potentially important) Lemma, an introductory sentence / paragraph could really help there. Similarly, stating what the bounds in Section 3.5 correspond to would also help understanding (especially given the fact that Observation 1 seems to be building on them).
  * Related to that, I find difficult to understand the exact link between the theoretical results and the numerical simulations, and consequently how the latter do help support / showcase the former. The experimental section seems to rely on Observation 1 only.


[1] Belinda Tzen and Maxim Raginsky. Theoretical guarantees for sampling and inference in generative models with latent diffusions. In Conference on Learning Theory, 2019

**Q5 Detailed Comments To The Authors:**

### Questions:
- **Abstract**, in what sense is "OU process scale better than PBM" meant? And how does that relate to the presented theoretical results ?
- **Intro** "As many theoretical works, … ". : it would maybe help to cite some of the works you refer to. And, could you comment on how restrictive that assumption is in practice?.
- **List of contributions**, why are Sections (3.4) and (3.5) not mentioned here?
- **Section 2**, the time reversals are said to be flipped, how is that ?— can’t seem to see it from the equations when comparing to DDS for instance.
- Right after **eq (4)**, isn't there a parenthesis missing in the LHS:  $\beta_{T-t}(...) =\beta_{T-t}y_t$?
- Right after **eq (8)**, what is $\theta^{*}$?
- **eq (9) - eq (10)**: How is alpha defined?;
- **Proposition 3.1** initial condition: what’s $x_0$ here? Shouldn't it be $\hat{x}_0$ instead. Also, why is it $p_1$ and not $p_0$?
- **section 3.3.1**, could you provide a word of explanation / intuition as to why it is called "terminal cost"?
- **Lemma 3.8**, what is $R$ in the envelope function?
- **section 3.5 title**, shouldn't the section be called *OU semigroup* instead *Heat semigroup*?
- **section 3.5, first paragraph** "The first bound is based on the derivations made earlier": which ones specifically?
- **Figure 1**, there is no reference to the figure in the text.
- **Section 4**: I am not sure that I understand what specific part of the theoretical results is surveyed with the numerical experiments. It seems to me that it is related to Observation 1, is that correctly understood?
- **Section 4.4**: "VP tends to sample points closer to the *initial distribution*" what is meant by *initial distribution* here?
- **Figure 4**, How should the ground truth look like? And also how can the funnel distribution be plotted in 2D when d>2, is it through some dimensionality reduction technique?
- What are the practical implications of the results presented in the paper?

### Readability:
 - "Page 1, second column, first paragraph" is a bit difficult to follow. Several shorter sentences would probably make things clearer.
 - "Page 2, first paragraph (starting with 'in this work')", the wording until the first ',' could be improved.
- References to proofs are sometimes missing from main text. For instance, Section 3.3. does not refer to Appendix C, as far as I can tell.

### Minor
- **General**: equations are not consistently encapsulated in the text, e.g. missing full stop (e.g. eq. (15)) or comma followed upper-case in the follow-up text (e.g. eq (16)).
- **abstract**: OU should maybe be made explicit at that stage;
- **section 2.3, first paragraph**:
- **Paragraph after eq (8)**: "The Equation (8)" -> "Equation (8)";
- **Section 2.4, last paragraph**: "whilst First ... to explore" -> "has explored";
- **Section 3.1**: "This process is commonly referred to ..." is missing a full stop at the end.
- **Remark 3.5**: "The time reversal of the VP-SDE" -> "The drift of the time reversal of the VP-SDE";
- `**In Lemma 3.8**: “to” -> “be”.
- **Proposition 3.1 and lemma 3.8**,  the $R$ should probably be blackboard $\mathbb{R}$ for domain and co-domain.
- **Section 3.4**: "Lemmas C.4" -> "Lemma".
- **Observation 1** is missing a word I believe: [...] entropy *that* corresponds to [...].
- **Page 7**: Koltchionskii -> Koltchinskii;
- **Appendix F**: PBN -> PBM

**Q9 Complying With Reviewing Instructions:**

Yes

---

> ### Author Rebuttal · Authors · 2024-04-02
>
> Dear Reviewer,
>
> We thank you for the insightful and incredibly thorough review. In what follows, we will address your questions and highlight changes to our manuscript that hopefully improve readability and address your concerns. As per the UAI email, we are allowed to share an anonymised link. Thus, we have uploaded our updated manuscripts (with your suggestions coloured in blue) to the following anonymous repository to aid the discussion.
>
> **Revised manuscript**:  https://anonymous.4open.science/r/rebuttal-A7C2/UAI.pdf
>
> Before addressing each question individually, we would like to address the broad weaknesses:
>
> > 1. without being an expert in the field, it seems to me that the novelty is limited as the paper consists in porting existing results from [1] to another class of SDEs, following a similar sketch for the different proofs (that being said, this does not mean that the paper does not constitute a valuable contribution);
>
> At a very high level, this is what we do. However, the main regularity properties required in Tzen and Raginsky to obtain a bound on the covering number of $\mathcal{G}$ no longer applies as $\bar{g}_{t,x}(z)$  is no longer Lipschitz in a Euclidean sense, instead its Lip wrt to the following metric (which we prove):
>
> $$\rho_{O U}\left((t, x),\left(t^{\prime}, x^{\prime}\right)\right)=|| e^{-t} x-x^{\prime} e^{-t^{\prime}} ||+\left|t-t^{\prime}\right|^{1 / 2}$$
>
>  As a result, we can no longer use the following properties of Euclidean-Lipschitz covers that is used in Tzen and Raginsky 2019 [1]:
>
> $$N\left(\mathcal{G}, L^2(Q), \varepsilon\|F\|_{L^2(Q)}\right) \leq N\left(\mathrm{~B}^d(R),\|\cdot\|, \varepsilon / 2\right) \cdot N\left([0,1],|\cdot|, \varepsilon^2 / 4\right)$$
>
> So, we must establish an akin property for our specific metric space  (see the proof for Lemma 3.10 in Appendix D).
>
> $$N\left([0, T] \times B^d(R), \rho_{O U}, \epsilon\right) \leq N\left([0, T],|\cdot|, \epsilon^2 / 4\right) N\left(B^d(R),\|\cdot\|, \epsilon / 2\right)$$
>
>  It was not obvious or straightforward; it required carefully constructing and combining several covers. This constitutes a non-trivial and new result that allows us to obtain covers for the function space induced by the OU-semigroup. Once this was obtained the general bounds were a bit different and thus a lot of symbol pushing plus some minor, easy derivations/manipulations were required to arrive at the final answer.
>
> Overall, we follow the high-level strategy from [1]; however, the road to there required several new results and a non-incremental (non-trivial) amount of work, which can be seen in our proofs.
>
> We have added a short paragraph in Section 3.3.1 where we briefly outline this contribution/difference.
>
> > 2. The experimental section seems to rely on Observation 1 only.
>
> This is accurate, from our specific results, our covering bounds lead us to Observation 1, which makes us conjecture that OU-based reversals are easier to “train” than PBM ones. We discuss this in more detail when addressing the reviewer's detailed questions.
>
> > 3. The presentation of the paper can be improved. I think that it could helpful to state in plain English what the theoretical results / derivations are / do, and what they mean.
>
> We have added general signposts and English/high-level descriptions of some of our main results; for example, see the paragraph just below Section 3.3.2:
>
> ```We must first obtain a notion of approximation "difficulty" in approximating the OU-semigroup to obtain estimation errors on the score. To do so, we must obtain bounds quantifying how many "tiles" (closed balls) are required to cover the space $\mathcal{G}$. The formal mechanism to do so is known as a covering number.```
>
> As there are many questions to address, we will answer them in separate official comments (as the system seems to allow for this? due to the discussion period).

---

### Meta-Review · Area_Chair_6Q4Z · 2024-04-16

The paper provides valuable theoretical analysis and insights on DDPMs which are of interest to the community. After author-reviewer feedback there is a consensus between the 4 reviewers for accepting based on extensive refinement of the presentation. Please make sure to implement all the suggestions and changes to the revised manuscript.